# Efficient Unsupervised Knowledge Distillation with Space Similarity

## Abstract

In this paper, we aim to boost performance of knowledge distillation without the ground-truth labels. Hence, a student can only rely on the response generated by its teacher. Many existing approaches under this setting rely on some form of feature/embedding queue to capture teacher's knowledge. These queues can be as large as over 100k samples. Also, these methods rely on multitude of operations which as a result increases the memory requirement for training many folds. In this work, we show that merely working with the input batch (often of size 256) it is possible to not only incorporate neighbourhood information but also obtain strong unsupervised distillation performance. We achieve this by introducing a simple space similarity loss component which works alongside the well known normalized cosine similarity computed on the final features. In this loss, we motivate each dimension of a student's feature space to be similar to the corresponding dimension of its teacher. With this seemingly simple addition, we are able to compete against many contemporary methods which either rely on large number of queued features or heavy pre-processing. We perform extensive experiments comparing our proposed approach to other state of the art methods on various computer vision tasks for established architectures. We will be sharing the official implementations to replicate our results and weights for the pre-trained models.

## 1 Introduction

Deep learning has achieved remarkable empirical success with the ever-increasing model sizes (Radford et al., 2019; Dai et al., 2021). However, these memory-intensive models pose a challenge when migrating their empirical success to smaller devices, which require lightweight and fast solutions. Furthermore, training a large network under a budget that requires significant computation may be infeasible for a wider audience. Nevertheless, larger networks have the advantage of being more capable of extracting meaningful information from data than their lightweight counterparts.

As a solution to the above mentioned scenario, knowledge distillation is proposed to transfer this rich knowledge from the larger network, referred to as "teacher", to a much smaller network, the "student" (Hinton et al., 2015; Buciluǎ et al., 2006). Hinton et al. (2015) extends the idea of model compression by Buciluǎ et al. (2006) for distilling knowledge of deep neural networks. The approach is task-specific and described for a supervised setting which relied on the task-specific output probability distribution over the target classes. Many methods since then

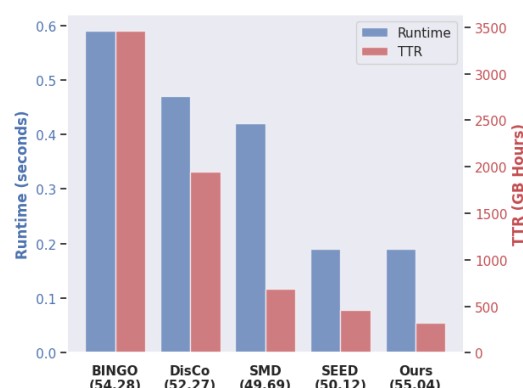

Figure 1: **Resources required to distill a ResNet-18 from a ResNet-50 teacher.** We report the runtime per iteration in seconds and the total training resources (TTR) required in GB hours. The evaluations were performed on a single DGX-A100 (80GB) GPU. Text annotations with the methods highlight the k-nearest classification accuracy (k=10) on ImageNet

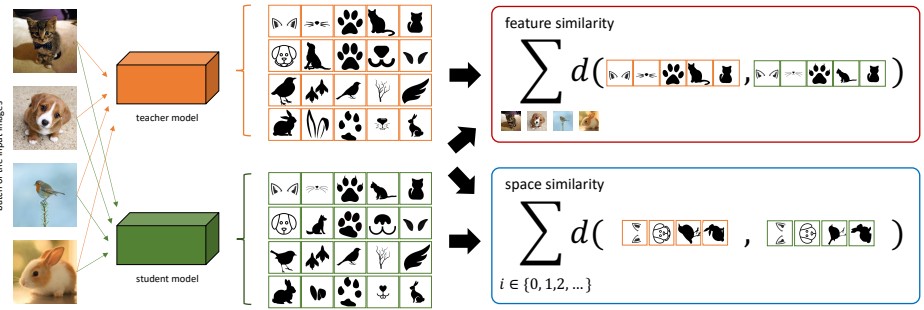

Figure 2: **The proposed CoSS distillation framework.** In the graphic, we demonstrate similarity of one pair of corresponding feature dimension being maximised. We perform this maximisation for every corresponding pair for the teacher and student.

have introduced different novelties to transfer knowledge via some form of divergence minimisation or similarity maximisation (Romero et al., 2014; Tung & Mori, 2019; Zagoruyko & Komodakis, 2017; Chen et al., 2021a). However, most of these methods require ground-truth labels during the distillation process.

In recent years, the development of self-supervised learning (SSL) has allowed networks to be trained on larger datasets without labels, leading to generic representations that are task agnostic and superior downstream performances once fine-tuned (Doersch et al., 2015; Misra & Maaten, 2020; Jing & Tian, 2021). As a result, SSL is an active area of reasearch. However, smaller networks do not readily benefit from SSL due to the smaller number of parameters, leading to the failure of these models in learning underlying discriminative representations (Fang et al., 2021). To address this issue, Fang et al. (2021) proposes an unsupervised knowledge distillation framework called SEED that allows smaller networks to take advantage of the large amount of data for pre-training. Since the introduction of SEED, many other approaches have followed suite (Xu et al., 2022; Koohpayegani et al., 2020).

A common theme amongst existing unsupervised knowledge distillation methods is that they

1. Employ contrastive loss, which aims to minimise the deviation between the student and teacher embeddings.
2. Utilize a large common embedding queue(s) against which similarity is computed for a training sample.
3. Batch sampling or pair generation to focus on 'important' samples.

Though these operations have proven to be beneficial, we found that they often require extensive amount of resources to function. From figure 1, it can be observed that the recent approaches of unsupervised distillation require a significantly large amount of computing. Moreover, as many approaches require multiple steps of processing and sample gathering, the training also becomes significantly slower. We found training these methods to be challenging, even with modest configurations. In comparison, our proposed method CoSS, consumes the least amount of GPU resources while also being considerably faster than its peers. This efficiency in distillation does not compromise the quality of the student, as is evident from the corresponding classification performance. CoSS requires roughly similar GPU to SEED but achieves $+5\%$ in accuracy. Moreover, compared to BINGO, CoSS is roughly $3\times$ faster and requires $3\times$ less GPU memory while providing comparable accuracy.

In this paper, we aim to perform unsupervised knowledge distillation while also keeping the required resources to a minimum. We believe that for distilling a small network, we should not require drastically large amount of resources. We hypothesize that, if the unnormalized embedding manifold of the student is similar to its teacher, the student has learnt its teacher's knowledge adequately. This would imply that student has learnt to map inputs in the 'same' way as the teacher onto the latent space. This will indirectly preserve various properties such as locality preservation, relative similarities which is often employed by existing distillation approaches. For this purpose, we propose a simple **space similarity** objective which works in conjunction with a traditional cosine similarity loss. In this loss, for each feature dimension of a student, we aim to maximise its similarity to the corresponding feature dimension of the teacher. Since this is processed over a training batch and only matching dimensions are considered, we manage to restrict the memory requirement for distillation

while also ensuring faster training. Figure 2 highlights our proposed loss component. Our main contributions are as follows:

- We introduce CoSS, a space similarity inclusive objective which motivates the student to mimic its teacher's embedding structure.
- Our simple method performs more efficient unsupervised distillation than other state-of-the-art methods.
- The simplicity of our approach does not impede the final performance of trained students. We report competitive results on standard distillation metrics.
- We evaluate various downstream tasks and find that the trained students with CoSS are more robust, transfers well, and are overall well-rounded.

The structure of the paper is as follows. In the subsequent section 2, we delve deeper into prior work. In section 3, we provide the details of our simple yet effective method. In section 4, utilizing a toy experiment, we demonstrate the importance of space alignment. In section 5, we report results on various benchmarks. We discuss the implications of our work and findings in section 6. Lastly, we conclude with final remarks in section 7.

## 2 RELATED WORK

Our study in this paper focuses on **unsupervised** distillation. Before we detail related works in the unsupervised regime, we first provide an introduction to the supervised distillation regime, which has a longer history, to better contextualize our proposed method.

### 2.1 SUPERVISED DISTILLATION

Early solutions for distillation have been designed for a fully supervised environment. Soft-label distillation (Hinton et al., 2015) is amongst the first work towards training smaller networks with guidance from larger teacher networks. Apart from supervised loss from the ground-truth labels, it minimises cross-entropy between the teacher's and the student's output distribution. The method relies on the teacher to be trained on a specific task and hence is dependent on a teacher's training methodology. Many subsequent approaches since then have incorporated internal layers, attention mechanisms etc. to match various novel objectives between student and the teacher (Romero et al., 2014; Zagoruyko & Komodakis, 2017; Yim et al., 2017; Huang & Wang, 2017; Kim et al., 2018; Koratana et al., 2019; Tung & Mori, 2019; Ahn et al., 2019). However, as most of these approaches require careful selection of an appropriate statistic, it can be a drawback in practice for defining the distillation procedure for newer architectures.

Another set of methods exploit local and global relationships (Yu et al., 2019; Peng et al., 2019; Park et al., 2019). They mostly differ in the relationships for optimisation: PKT (Passalis & Tefas, 2018) models the interactions between the data samples in the feature space as a probability distribution. RKD (Park et al., 2019) enforces the student to learn similar distances and angles between the training samples to that of its teacher. DarkRank (Chen et al., 2018) employs learning to rank framework (Cao et al., 2007; Xia et al., 2008) for distillation with a focus on metric learning. LP (Chen et al., 2021a) further develops the idea of FitNets and introduces locality-preserving loss, which relies on identifying K-nearest neighbours within the training batch. Huang et al. (2022) introduces a correlation based formulation which is very similar to ours. The key difference apart from the problem setting (supervised vs. unsupervised) is that they normalise the predictions (via. softmax) and then compute inter and intra-class similarities afterwards. Whereas, we independently normalize spatial and feature dimensions. From the perspective of computing intra-class similarity, it is logical to apply the softmax beforehand for generating class-wise scores, however, when operating on the embedding space, any normalisation on the features alters the space information as well.

There are also methods which utilize self-supervision for knowledge distillation. CRD (Tian et al., 2020), WKD (Chen et al., 2021b) and SSKD (Xu et al., 2020) fall into this category. Specific to the vision transformers, DeiT (Touvron et al., 2021) is a fully supervised distillation method that uses distillation tokens. Distillation token plays a similar role as a class token, except that it aims at reproducing the label estimated by the teacher. Other approaches such as Coadvice(Ren et al., 2022) and DeadKD (Chen et al., 2022) focus on the data inefficiency aspect of supervised distillation. Lastly, there are also approaches which perform softmax regression (Yang et al., 2021), neural architecture search(Dong et al., 2023), and two stage training with novel modules(Liu et al., 2023).

## 2.2 Unsupervised Distillation

SEED (Fang et al., 2021) is the first work, to the best of our knowledge, that attempts to distill knowledge in a purely unsupervised fashion. They perform knowledge distillation of a self-supervised teacher by minimizing the divergence between the similarity response of teacher and student on a **common embedding queue**. This approach is similar to the supervised method of SSKD (Xu et al., 2020), but instead of the mini-batch, a much larger queue is used. CompRes (Koohpayegani et al., 2020) introduces two features queues for each teacher and student. DisCo (Gao et al., 2022) performs a consistency regularization between augmented versions of the input in addition to unsupervised distillation. BINGO (Xu et al., 2022) is a two-stage method for performing unsupervised distillation, and both stages are resource intensive in terms of memory and time. Also, like many other approaches, it also relies on a **large feature queue** with a relatively heavier augmentation in the form of CutMix(Yun et al., 2019). SMD (Liu & Ye, 2022) focuses on mining hard positive and negative pairs rather than operating on all pairs for distillation. To counter the influence of wrongly assigning positive and negative labels, they utilized a weighting strategy. Though they did not use a feature queue, the costly operations of finding hard samples and learning the weighting strategy added to the resource overhead.

## 2.3 Key Differences

Our proposed method is designed for unsupervised distillation, but we believe it stands out from existing methods in both supervised and unsupervised regimes, despite its simplicity. In particular, our approach focuses on directly motivating the student to learn its teacher's latent manifold.. As a quantitative summary, CoSS differs from existing methods in the absence of (i) feature queues, (ii) contrastive objectives, (iii) heavy augmentations, and (iv) and custom batch composition (sampling strategy).

## 3 Method

We first introduce notations and then explain the CoSS objective. Let $f_t$ and $f_s$ be the teacher and student deep neural networks respectively. Since, we are working in an unsupervised setting, the labels are not known. We consider a dataset $\mathcal{D} = \{x_0, x_1 \ldots x_n\}$ consisting of $n$ data points. The embeddings $f_t(x) \in \mathcal{R}^{d_t}$ and $f_s(x) \in \mathcal{R}^{d_s}$ is the response gathered at the penultimate layer of a network (typically after global average pooling (Lin et al., 2014a)). The scenario where $d_s \neq d_t$, we can use a small projection head for the student which can then be discarded after the distillation process (Fang et al., 2021; Xu et al., 2022). We thus simply replace $d_t$ and $d_s$ by $d$. Let $B$ be the current mini-batch in training of size $b$. We can denote the embedding representations of all inputs in $B$ generated by the teacher and student as matrices $A_t$ and $A_s \in R^{b \times d}$ respectively. Here, $A_*^i$ is the embedding output $f_*(x_i)$.

Let, $\hat{}$ denote a L2 normalized vector. We can then compose a matrix of normalized feature vectors $\hat{A}_* = [\hat{A}_*^0, \hat{A}_*^1, \ldots \hat{A}_*^b]^T$. The widely known normalized cosine similarity loss on features is defined as:

$$\mathcal{L}_{co} = -\frac{1}{b} \sum_{i=0}^{b} cosine(\hat{A}_s^i, \ \hat{A}_t^i) \tag{1}$$

The loss computes the cosine similarity between corresponding embeddings of the teacher and student. Similarly, we can also compute similarity for each feature dimension. We define transpose matrix of features $Z_* = A_*^T$ and its normalized version as $\hat{Z}_*$. The space similarity component of our final loss is

$$\mathcal{L}_{ss} = -\frac{1}{d} \sum_{i=0}^{d} cosine(\hat{Z}_s^i, \ \hat{Z}_t^i). \tag{2}$$

It is very simple to implement and only requires one to transpose the feature matrices prior to the similarity computation. Our final objective is composed of weighted combination of **Co**sine similarity and **S**pace **S**imilarity:

$$\mathcal{L}_{CoSS} = \mathcal{L}_{co} + \lambda \mathcal{L}_{ss} \tag{3}$$

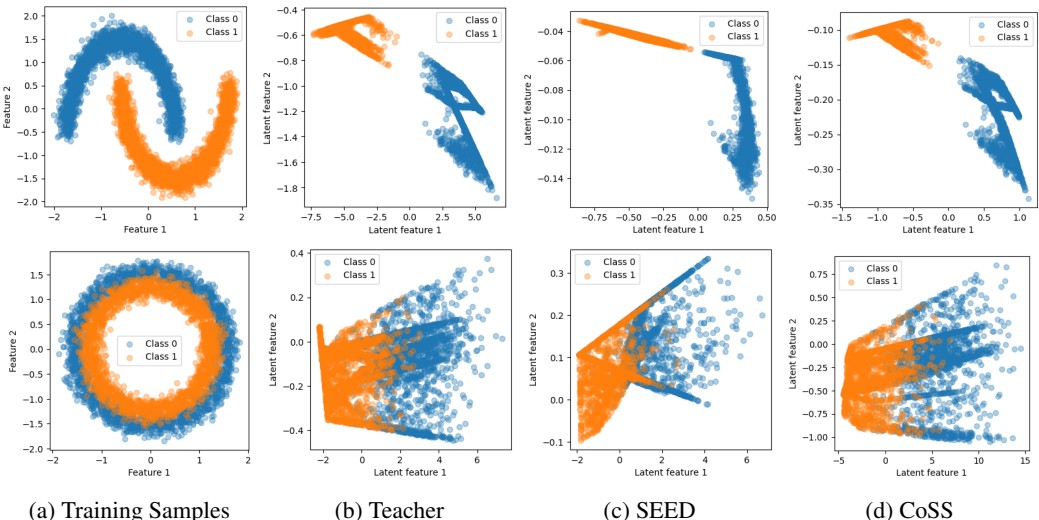

| (a) Training Samples | (b) Teacher | (c) SEED | (d) CoSS |

Figure 3: **Plots comparing the latent space of the teacher with SEED and CoSS students.** Eventhough the SEED student is able to learn a viable solution (achieving comparable downstream accuracy), the latent space learnt is significantly different from its teacher. CoSS, on the other hand, is able to faithfully model it's teacher's embedding space better.

Figure 2 provides a pictorial representation of our approach. Feature similarity compares the features being extracted for a single sample by the teacher and student whereas, space similarity compares features from different input samples together.

CoSS only performs pairwise comparison which is one of the main differences from existing unsupervised distillation methods which leverage a contrastive learning framework and perform one-to-many comparisons. This pair is formed only of "positive" samples, i.e. the output embeddings for the same input for teacher and student for $L_{co}$ and corresponding spatial dimensions of teacher and student for $L_{ss}$. Pairwise alignment or considering only positives has been explored successfully in the setting of self-supervised learning (Grill et al., 2020; Chen & He, 2021) which we have applied to the problem of unsupervised distillation.

## 4  IMPORTANCE OF SPACE SIMILARITY

In deep learning, it is generally accepted that neural networks learn a mapping from high dimensional space to a low dimensional manifold. However, explicit statements about the assumption of a locally euclidean manifold in the context of unsupervised learning can be relatively rare, as this assumption is often fundamental to the underlying methods without being explicitly articulated in the papers. For example, many unsupervised learning methods employ manifold learning based data visualisations(Oord et al., 2018; Zhuang et al., 2021). These manifold learning approaches in their workings assume the embedding manifold to be locally eucldiean(Cayton et al., 2008; Van der Maaten & Hinton, 2008). This assumption of the manifold being locally euclidean, allows us to treat the embedded manifold as a topological manifold. Here, we present a simple argument to show that methods only relying on an embedding queue cannot learn the teacher's embedding structure reliably.

**Definition 4.1.** *Two topological spaces $\mathcal{X}$, $\mathcal{Y}$ are homeomorphic if there exists a mapping $f : \mathcal{X} \rightarrow \mathcal{Y}$ s.t. $f$ is continuous, bijective and its inverse function $f^{-1}$ is also continuous.*

Homeomorphism(Poincaré, 2010) defines the concept of similarity (equivalence) between the two topological spaces. For SEED (Fang et al., 2021) and similar others, which only rely on normalized cosine similarity, the student's normalized manifold and the original teacher's manifold are not homeomorphic. This is because the operation of $L_2$ normalisation is not a homeomorphism. It is not continuous, bijective, and lacks a continuous inverse. A straight forward example to support it is that, all points lying on the ray starting from the origin will result in multiple points being mapped onto the same point on a $d-$dimensional hyper-sphere. Hence, minimisation of an objective operating on the normalized space will not preserve the original **un-normalized** structure. With space similarity, for a point, dimension-wise normalization (or scaling down) is a bijective and continuous mapping with a differentiable inverse. This implies that dimension normalisation preserves the similarity between

unnormalized and normalized manifolds. With this added objective, we aim to approximate the normalized embedding space of a teacher thereby, also approximating the original un-normalized space.

With figure 3, we demonstrate our argument with a set of toy experiments (training details in section A.1). Using simple datasets, we first train the teacher for a classification task. It can be observed that the teacher's embedding manifold has a non-trivial structure associated with it. Next, using SEED and CoSS' objectives, we perform the unsupervised knowledge distillation. SEED, though able to produce linearly separable latent embeddings, fails to adequately replicate teacher's manifold. However, with CoSS, we are able to replicate finer structures in the teacher's latent space. In Appendix A.3, we quantitatively measure goodness of CoSS' learnt latent embedding space. Specifically, we measure the similarity of local neighbourhood between a student and its teacher.

## 5 EXPERIMENTS

### 5.1 UNSUPERVISED DISTILLATION

We follow SEED's protocols for training and distilling the student networks. For SMD, however, we followed the authors' reported training recipe which was different to that of SEED. We compare our work with existing unsupervised distillation approaches of SEED, BINGO, DisCo, and SMD. We also include the extension of standard KD objective (Hinton et al., 2015) to the feature space and denote is as SKD. Previously, this extension has been applied for self-supervised learning (Caron et al., 2021).Further implementation details are provided in the Appendix (section A.1).

**Linear evaluation:** We train the classification layer while freezing the remainder of the student network. In table 1, we report the top-1 accuracy computed over the ImageNet validation set. CoSS consistently outperforms SEED while providing competing performance to resource hungry BINGO and DisCo on ResNet-18 and Efficientnet-b0. SKD, where we applied the standard KD in the feature space performs well on the unsupervised distillation task. It is also interesting to note that for ViT, there is a significant drop in performance of baselines. Often, a drop in performance is observed during the cross-architecture distillation (Tian et al., 2020). The reason given for such an observation is the difference in the inductive biases of the networks. We suspect that in addition to this, our alignment based objective is less constraining than the contrastive loss based objectives of the baselines.

**KNN classification:** Linear evaluation introduces additional parameters to the benchmarking. It often requires extensive hyper-parameter tuning to find optimal parameters. K-nearest classification on the other hand, allows us to evaluate the methods directly without the need of tuning parameters (Xu et al., 2022; Fang et al., 2021). As a result, we believe KNN is a fairer evaluation strategy for comparing different distillation methods.

We choose 10 nearest neighbours from the training samples and assign the majority class as the prediction (ablation in section A.2 provides results on different neighbourhood size). As shown in table 1, CoSS achieves state of the art performance for this metric. Also, it is interesting to note that even though for scenarios where CoSS performed lower in top-1, we achieved better KNN accuracy. We believe that this is due to the better modelling of the manifold which the space similarity objective provides.

### 5.2 EFFICIENCY

A main goal for distillation is to train a smaller netowork which is on par with its teacher. We believe in-order to distill onto a small student the resources required should not be drastically large. Here, we show that our approach requires the least amount of resources to perform unsupervised distillation. Mainly, we measure and report training time per iteration (in seconds) computed over 1000 iterations and Total-training-resources (TTR) required to train the student model. We compute TTR as the product of total number of training iterations, mean peak GPU memory usage per iteration and total training time of the distillation process. It captures the total amount of memory consumed for the entirety of training (reported in GB Hours). For a fair comparison, we computed each method on identical machines (Nvidia-A100 80GB) with auto-mixed precision disabled where necessary and set the batch size as 256.

Table 1: **Unsupervised distillation of a (self-supervised) teacher on ImageNet-1K.** The *teacher* and *student* correspond to models trained using Moco-v2. Values with **\*** are reported directly from the original paper. DisCo training fails for the ViT and hence the missing entry. We highlight the best student performance in **bold**

| Methods | ResNet-18 | | ResNet-50 (67.40) ResNet-34 | | Eff-b0 | | ViT-Ti | |
|---|---|---|---|---|---|---|---|---|
| | Top-1 | KNN-10 | Top-1 | KNN-10 | Top-1 | KNN-10 | Top-1 | KNN-10 |
| Student | 51.38 | 41.99 | 57.40 | 41.50 | 41.81 | 31.10 | 27.70 | 23.27 |
| SKD (Hinton et al., 2015) | 58.16 | 54.31 | – | – | 61.49 | 56.10 | – | – |
| SEED (Fang et al., 2021) | 57.96 | 50.12 | 58.50* | 45.20* | 61.44 | 53.11 | 51.93 | 42.15 |
| BINGO (Xu et al., 2022) | **60.10** | 54.28 | **63.50*** | × | 62.01 | 54.75 | 48.67 | 51.14 |
| DisCo (Gao et al., 2022) | 60.01 | 52.27 | 61.74 | 53.65 | **63.95** | 54.78 | – | – |
| SMD (Liu & Ye, 2022) | 59.56 | 49.69 | – | – | – | – | – | – |
| CoSS (ours) | 59.24 | **55.04** | 60.95 | **55.61** | 63.55 | **58.32** | **61.75** | **57.96** |

We present our findings in table 2. SEED, SKD, and CoSS, which rely on simpler objectives require the least amount of resources for training. The higher TTR for SEED is the result of extra memory consumed in form of a feature queue. BINGO, consumes the most amount of resources owing to its bagging datasets and sampling strategies. TTR on SMD is relatively lower even though it is comparable to DisCo on runtime is because it is trained for 100 epochs instead of the 200 which was used for DisCo.

Table 2: **Comparison of run-time efficiencies of different methods.** We report the mean time (seconds) per iteration and total-training resources. Lower is better

| Method | Runtime (seconds) | | TTR (GB Hours) | |
|---|---|---|---|---|
| | ResNet-18 | Eff-b0 | ResNet-18 | Eff-b0 |
| SKD | **0.19** | **0.41** | **322.26** | **2745.18** |
| SEED | **0.19** | **0.41** | 463.32 | 3035.89 |
| BINGO | 0.59 | 1.26 | 3461.51 | 26,195.66 |
| DisCo | 0.47 | 0.82 | 1945.91 | 11,174.55 |
| SMD | 0.42 | 0.98 | 685.51 | 6492.18 |
| CoSS | **0.19** | **0.41** | **322.26** | **2745.18** |

## 5.3 TRANSFER LEARNING

To understand the transferability of the learnt students, we perform the benchmark using the public implementation of Ericsson et al. (2021). Under this evaluation, we freeze the backbone and only learn the classification layer on top of the network. To make the evaluation fair, we perform a hyper-parameter sweep across numerous values to identify the best set of values for each model for every dataset. We report the top-1 classification from the identified best setting.

As it can be observed from table 14, the classification accuracy is highest for CoSS across various datasets. Previously, we had noticed that baseline methods performed well on linear evaluation for ImageNet. However, we now observe that under a fairer hyper-parameter search based linear evaluation, CoSS achieves higher performance for ResNet-18 and Efficientnet-b0 reflecting the good quality of learnt features.

Table 3: **Transfer learning evaluation of distilled ResNet-18 and Efficientnet-b0.** Here, we report the top-1 accuracy. In the appendix, we report the KNN accuracy of the models.

| Method | CIFAR-10 | | CIFAR-100 | | STL-10 | | Caltech-101 | | Pets | | Flowers | | DTD | |
|---|---|---|---|---|---|---|---|---|---|---|---|---|---|---|
| | ResNet-18 | Eff-b0 | ResNet-18 | Eff-b0 | ResNet-18 | Eff-b0 | ResNet-18 | Eff-b0 | ResNet-18 | Eff-b0 | ResNet-18 | Eff-b0 | ResNet-18 | Eff-b0 |
| SKD | 88.97 | – | 70.12 | – | 93.80 | – | 86.90 | – | 78.31 | – | **88.60** | – | 72.34 | – |
| SEED | 85.27 | 88.85 | 62.75 | 69.87 | 93.99 | 94.90 | 80.26 | 84.98 | 76.18 | 78.81 | 75.10 | 88.44 | 67.34 | 69.79 |
| BINGO | 87.67 | 89.74 | 66.14 | 70.25 | 94.99 | 94.75 | 83.84 | 86.48 | 79.24 | 80.80 | 83.62 | 90.35 | 70.00 | 70.85 |
| DisCo | 88.11 | 91.63 | 67.50 | 73.97 | 95.04 | **95.78** | 84.65 | 86.44 | 77.86 | 81.20 | 83.69 | 89.52 | 69.89 | 71.91 |
| SMD | 86.47 | – | 64.42 | – | 94.24 | – | 80.59 | – | 74.59 | – | 78.97 | – | 69.31 | – |
| CoSS (ours) | **89.84** | **92.51** | **70.03** | **76.72** | **95.31** | 95.41 | **87.06** | **89.26** | **80.31** | **82.42** | 87.04 | **92.57** | **71.54** | **74.84** |

## 5.4 ROBUSTNESS

Various studies have indicated that deep learning systems often break down when encountered with data outside the training distribution(Hendrycks & Dietterich, 2019; Geirhos et al., 2019; Wang et al., 2019). Due to the wide spread applicability of deep learning systems in the real world, it becomes important to ensure a high degree of robustness of such systems. In this experiment we explore the robustness of trained students under different kinds of shifts in the input data. ImageNet-v2 is a natural image dataset which closely resembles the sampling process of the original ImageNet dataset. It has been employed by various studies to understand the robustness of models under natural data shift (Taori et al., 2020; Ovadia et al., 2019). ImageNet-S consists of sketch images for ImageNet classes. It is a stronger deviation from the types of images present in the training set. ImageNet-C is a synthetically generated dataset for evaluating a model's performance under various forms of corruptions. We utilised corruption=1 for our evaluation.

Results reported in table 4 demonstrate that CoSS is robust across various input distribution shifts. It even outperforms BINGO which employs stronger data augmentations in the form of CutMix. Strong data augmentations such as CutMix have been shown to improve the robustness.

Table 4: **Robustness evaluation of ImageNet distilled ResNet18s.** To emphasize on the quality of embeddings learnt, we report the KNN-10 accuracy. We highlight the best performance in **bold**. We report the robustness evaluation of efficientnet-b0 in the appendix

| Method | ImageNet-v2 | | | ImageNet-S | ImageNet-C | | | | | | | | | | | | | | |
| | MF | Tr | Top | | brightness | contrast | defocus | elastic | fog | frost | gaussian | glass | impulse | jpeg | motion | pixelate | shot | snow | zoom |
| SEED | 37.58 | 45.11 | 51.20 | 10.41 | 47.32 | 42.38 | 32.20 | 43.01 | 32.47 | 34.98 | 35.26 | 30.58 | 20.35 | 34.51 | 35.34 | 41.35 | 33.26 | 27.73 | 24.68 |
| BINGO | 41.01 | 48.99 | 55.03 | 12.78 | 51.40 | 47.02 | 36.16 | 47.17 | 37.20 | 39.44 | 37.57 | 33.67 | 20.25 | 40.60 | 37.46 | 46.14 | 36.29 | **31.28** | 27.02 |
| DisCo | 39.49 | 46.96 | 53.18 | 12.37 | 49.89 | 45.61 | 35.54 | 45.60 | 35.69 | 38.11 | 37.37 | 33.90 | 21.98 | 38.39 | 38.79 | 45.06 | 36.28 | 30.31 | 26.77 |
| SMD | 37.86 | 44.45 | 50.83 | 10.27 | 47.06 | 42.78 | 32.89 | 42.92 | 32.46 | 34.04 | 33.17 | 30.39 | 17.87 | 35.48 | 34.69 | 41.64 | 31.67 | 26.68 | 24.74 |
| CoSS (ours) | **41.74** | **49.10** | **55.44** | **12.85** | **52.65** | **49.03** | **39.02** | **48.12** | **38.40** | **39.57** | **40.27** | **35.94** | **23.56** | **43.49** | **40.82** | **48.52** | **37.98** | 31.12 | **28.68** |

## 5.5 IMAGE SEGMENTATION

We evaluate trained students under two settings (more details in A.1) 1. Traditional image semgentation, where we train a FCN(Long et al., 2015) head on top of the student. We evaluate this model on CityScapes(Cordts et al., 2016) and CamVid(Brostow et al., 2009) datasets. 2. Following Caron et al. (2021), we employ the self attention maps and threshold it to obtain segmentation masks. We evaluate the maps on the PascalVOC dataset(Everingham et al., 2015), MSRA(Liu et al., 2007; Hou et al., 2019) and ECSSD(Yan et al., 2013).

For traditional image segmentation, as reported in 5, we can observe that the CoSS trained student serves as an improved backbone over the baselines. For the segmentation evaluation using the attention maps, we found that BINGO ignores [CLS] token after a certain depth (6 in this case). Our probing indicated that towards the end of the network, BINGO distilled ViT-tiny gave equal attention to all input patches which resulted in poor image segmentation performance. We report the [CLS] self-attention maps evaluated prior to the degradation using $^+$. The results indicate that the quality of attention maps generated by CoSS is superior to those from the baselines. We'd like to emphasize that for this evaluation, no additional training is performed. We provide visualisations of the self-attention maps in figure 5.

Table 5: **Image segmentation evaluation**. Higher is better.

| Method | CamVid | | Cityscapes | | VOC | MSR | ECSSD |
| | $Acc_p$ | $IoU_m$ | $Acc_p$ | $IoU_m$ | $IoU_m$ | $IoU_m$ | $IoU_m$ |
| SEED | 86.17 | 0.2573 | 82.53 | 0.2867 | 34.01 | 34.30 | 29.96 |
| BINGO | 87.43 | 0.2761 | 83.64 | 0.3099 | 37.31$^+$ | 41.34$^+$ | 43.11$^+$ |
| DisCo | 86.91 | 0.2791 | 82.85 | 0.2933 | – | – | – |
| SMD | 75.67 | 0.1620 | 82.38 | 0.2774 | – | – | – |
| CoSS | **88.00** | **0.2855** | **84.39** | **0.3115** | **42.28** | **47.71** | **49.84** |

Table 6: **DAVIS 2017 Video object segmentation**. Higher is better

| Method | $J_m$ | $J_r$ | $F_m$ | $F_r$ | $(J\&F)_m$ |
| --- | --- | --- | --- | --- | --- |
| SEED | 50.74 | 58.37 | 54.61 | 64.29 | 52.68 |
| BINGO$^+$ | 53.78 | 65.35 | 57.80 | 70.65 | 55.79 |
| CoSS | **56.07** | **66.98** | **59.07** | **71.32** | **57.58** |

## 5.6 VIDEO INSTANCE SEGMENTATION

This evaluation accesses the quality of frozen features on video instance tracking for imagenet distilled ViT-tiny networks. In table 6, following the experiment protocols of Jabri et al. (2020), we evaluate the self-attention maps on the DAVIS-2017 video instance segmentation task. In this evaluation, a scene is segmented by directly computing the nearest neighbours between consecutive

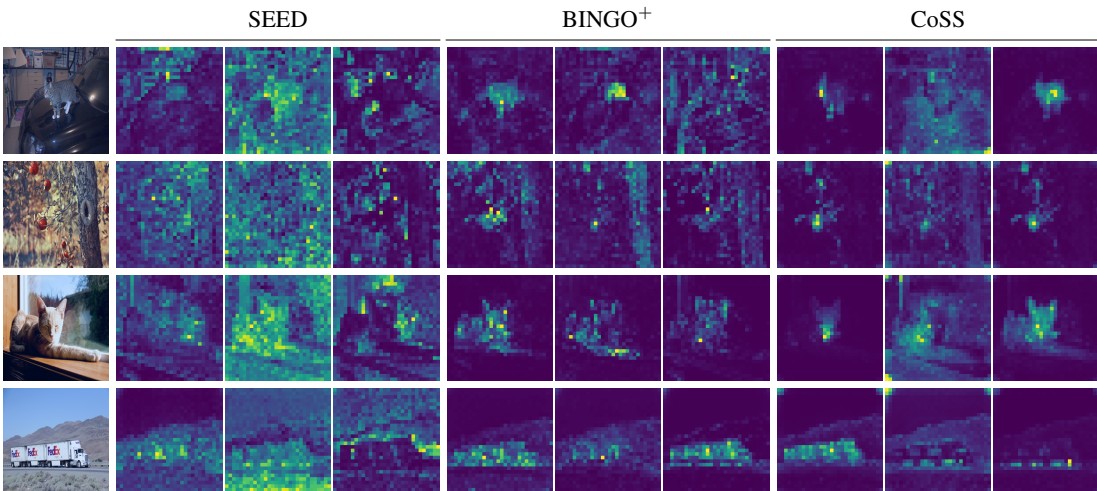

Figure 4: **Self-attention maps of `[CLS]` token from the** 3 **ViT-tiny heads.** We follow Caron et al. (2021) to visualize the attention maps. The images are randomly selected from COCO validation set, and are not used during training Lin et al. (2014b). The bright regions denote the patches of interest to the model. SEED's generated maps are relatively noisier and often deviate from the primary subject in the image. More visualisations are provided in the appendix

Table 7: **Top-1 accuracy of supervised methods adapted for unsupervised distillation**. We report the distillation performance of the methods without the contribution from the supervised losses. We also report the supervised performance as a baseline measure for both the student and the teacher

| Methods | ResNet56(0.86M) ResNet20(0.27M) | VGG-13(9.45M) VGG-8(3.96M) | ResNext32x4(7.43M) ResNext8x4(1.23M) | WRN-40-2(2.25M) WRN-16-2(0.7M) | ConvMixer-256/12(0.95M) ConvMixer-256/4(0.19M) | ViT-384/12/7(6.26M) ViT-384/4/3(2.7M) |
|---|---|---|---|---|---|---|
| Teacher | 72.41 | 74.64 | 79.42 | 75.61 | 73.45 | 66.70 |
| Student | 69.10 | 70.31 | 71.79 | 73.01 | 68.02 | 63.52 |
| ATT (Zagoruyko & Komodakis, 2017) | 49.00 | 56.22 | 43.11 | – | – | – |
| SP (Tung & Mori, 2019) | 63.14 | 71.62 | 62.51 | – | – | – |
| VID (Ahn et al., 2019) | 68.61 | 73.65 | 69.60 | – | – | – |
| RKD (Park et al., 2019) | 56.28 | 44.50 | 32.33 | 52.38 | 46.41 | 28.17 |
| PKT (Passalis & Tefas, 2018) | 61.58 | 71.81 | 64.53 | 70.75 | 53.82 | 29.64 |
| Factor (Kim et al., 2018) | 46.63 | 39.71 | 36.64 | – | – | – |
| NST (Huang & Wang, 2017) | 25.62 | 42.78 | 41.27 | – | – | – |
| CRD (Tian et al., 2020) | 65.03 | 69.73 | 65.78 | 70.62 | 61.89 | 48.70 |
| SRRL (Yang et al., 2021) | 69.31 | 72.85 | 69.19 | 73.44 | – | – |
| DIST (Huang et al., 2022) | 67.13 | 73.40 | 67.67 | 72.85 | – | – |
| SKD | 68.69 | 73.33 | 72.52 | 73.01 | – | – |
| CoSS (Ours) | **71.11** | **74.58** | **73.90** | **74.65** | **72.73** | **63.31** |

frames utilizing the features from different depths of the frozen network. We report mean region similarity $J_m$, region similarity recall $J_r$, mean contour-based accuracy $F_m$ and contour-based recall $F_r$. As the results indicate, the features learnt by CoSS are able to retain better spatial information than the baselines.

## 5.7 (UN)SUPERVISED DISTILLATION

In previous experiments we have compared CoSS against methods designed for unsupervised distillation. In section 2, we discussed about supervised distillation methods. A key point to note about majority of supervised methods is that even though they were proposed for a supervised setting, often, the key novelty lied in an unsupervised component. In this section, we adapt and benchmark existing supervised distillation methods for the task of unsupervised distillation.

**Methodology:** Apart from using the batch size of 256, we strictly follow the benchmark evaluation protocol of CRD(Tian et al., 2020) on CIFAR-100 for this task. The teachers are trained with supervision and for the process of distillation we remove the final classification layer to simulate an unsupervised setting. In recent architectures, for methods which rely on intermediate layer, we select the layers midway across the depth of a network.

**Results:** As shown in table 7, many supervised methods don't readily adapt to the unsupervised scenario. They either fail completely or they fail to provide any adequate performance. CoSS due to its simplicity can be employed by any teacher-student pair.

## 6    DISCUSSION

Unsupervised knowledge distillation is an important problem that we have addressed in this work. Existing solutions to this problem require significant computing and training time. For smaller networks that can be conventionally trained on smaller GPUs, it becomes hard to justify the need of deploying GPU clusters to perform distillation. CoSS provides the benefit of faster and more efficient distillation without compromising on the final performance.

Objective of CoSS only relies on positive pairs of samples, this is a prominent difference from contrastive learning baselines. Tian et al. (2021) study various factors which impact training with only positive samples in the context of SSL. We observed that unlike in an SSL training, for an unsupervised distillation, predictor/projection was not a hard requirement. CIFAR-100 experiments were performed in the absence of a predictor head and the distillation concluded without any difficulties. This leads us to believe that observations which hold true for an online SSL approach may not hold true for its adaptation for an unsupervised distillation. Subsequently, this direction warrants further investigation in the future to elucidate any potential relationships.

Furthermore, rather than distilling only similarity assisted by feature queues, we directly focused on approximating the learnt manifold using space similarity. Toy experiments and analysis of common embedding neighbours supports our intuition. The local neighbourhood is preserved even in comparison to BINGO, which applies an explicit KNN over the entire training data. In section B, we demonstrate that, in theory we can utilise BatchNorm (Ioffe & Szegedy, 2015) to model teacher's embedding manifold. We believe better approaches to latent space modelling will improve the distillation performance further.

For large-scale datasets like ImageNet, we found that CoSS often performed competitively compared to GPU-intensive methods. However, a KNN-based evaluation showed that embeddings learnt by CoSS are indeed meaningful and better. This performance is also maintained across a wide range of nearest neighbour sizes. Recent studies have also advocated reliance on KNN accuracy for drawing meaningful conclusions (Fang et al., 2021; Caron et al., 2021). Furthermore, qualitative analysis of the tasks of transfer learning and robustness further strengthens our argument.

We also reported cross-architecture distillation results on ViT-tiny. We observed difficulty in utilising existing methods for ViT. For DisCo, the training fails arbitrarily as the loss value becomes undefined. For BINGO, the distillation process ignored the [CLS] token in the model after a certain depth. Compared to the baselines, CoSS operates as intended and is able to produce a fully functional student ViT-tiny.

We found that applying existing (supervised) methods to newer architectures in an unsupervised setting does not extend well. They either fail to train or underperform. This hints towards high dependency on data labels for these methods to work effectively. Moreover, many of these methods rely on defining correspondences between intermediate layers. Defining such a mapping is non-trivial and requires a number of trials and errors. Recalling that in a supervised setting, the novelty often lies in an unsupervised component and due to the simplicity of CoSS, we believe that our approach can easily be extended to a supervised setting.

Lastly, due to the simplicity of our approach, we believe that it can also be extended into the problem settings of cross domain distillation (Zhang et al., 2021), distillation of joint vision-language models, cross lingual distillation(Reimers & Gurevych, 2020; Gupta et al., 2023). We aim to explore these avenues in the future.

## 7    CONCLUSION

In this paper, inspired by the need that large models trained with self-supervised learning will also be considered to distilled into smaller models, but the labels used to train these large models are usually not available, we investigated the knowledge distillation in a pure unsupervised setting.

In this setting, we showed that unsupervised feature distillation can be performed without storing a feature queue. To enable this, we proposed CoSS, a feature and space similarity loss. We demonstrated that the proposed method is able to retain the local neighbourhood similar to that of its teacher. Moreover, across various benchmarks, it achieves better performance than the existing state-of-the-art methods. Focusing on Vision Transformers, we demonstrated the high quality of attention maps being learnt through our approach with an added benefit of a stable training.

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

# A  APPENDIX

## A.1  IMPLEMENTATION DETAILS

**Teacher model:** Following SEED, we utilize the Moco_v2 trained ResNet-50 as the teacher in our experiments. We discard the projection head of the teacher and work directly with the 2048 dimensional features from ResNet-50. We utilized their official implementation in our ImageNet based experiments.

**Student models:** We employ a mix of established and recent architectures in our study. We perform distillation on ResNet-18(He et al., 2016a), Efficientnet-b0(Tan & Le, 2019), ViT-tiny(Dosovitskiy et al., 2021). The student models are trained from scratch. To match the final representation sizes, similar to DisCo, we add a 2-layered MLP on top of the student network which is discarded after distillation. For training on CIFAR-100, ResNet56/20, ResNet32x4/ResNet8x4, VGG13/8 and WRN_40_2/WRN_16_2 are detailed in CRD. For the Convmixer, we set it embedding dimension as 256, depth 12 for teacher and 4 for the student, and kernel size of 7 and patch size of 1. For ViT, we set the embedding dimension as 384, num_heads=12 for teacher and 4 for the student, and number of layers 7 for the teacher and 3 for the student. We would like to note that architectures for convmixer and ViT maybe not be optimal for CIFAR-100 and there may exist better configurations.

**Training:** For training, we use the configuration of SEED. Under this setting, the distillation process is performed for 200 epochs with a batch size of 256. The starting learning rate is $0.03$ which is slowly reduced to $0$ using a cosine scheduler over the course of training. SGD is used as the optimizer with momentum $(0.9)$ and weight decay $(0.0001)$. For CoSS, we set $\lambda = 0.5$. For the baselines, we utilize their corresponding official implementations. For more detail please refer SEED(Fang et al., 2021). We reproduced SMD's results using their official source code and hyper-parameters. For SKD, we used the temperature scaling parameters for student and teacher as $\tau_s = 0.1$ and $\tau_t = 0.07$ respectively as per Caron et al. (2021). For SKD, we also used a loss scaling parameter $\beta = 100.0$.

**Datasets:** We utilize a large collection of datasets in our work. We primarily use ImageNet (Deng et al., 2009) for performing unsupervised distillation. To evaluate the quality of representations learnt, we perform transfer learning on CIFARs(Krizhevsky, 2009), STL-10 (Coates et al., 2011), Caltech-101 (Fei-Fei et al., 2006), Oxford-IIIT Pets (Parkhi et al., 2012), Flowers (Nilsback & Zisserman, 2006) and DTD (Cimpoi et al., 2014). We also utilize various ImageNet variants namely, ImageNet-v2 (Recht et al., 2019), ImageNet-Sketch (Wang et al., 2019) and ImageNet-C (Hendrycks & Dietterich, 2019) to understand robustness of distilled networks. For visualization of attention maps and their quantitative analysis, we utilize COCO (Lin et al., 2014b), PascalVOC (Everingham et al., 2015), MSRA(Liu et al., 2007; Hou et al., 2019), ECSSDYan et al. (2013) DAVIS (Pont-Tuset et al., 2017).

**Toy Experiment:** For the toy experiment we use a 5 layered MLP with $[4, 8, 4, 2, 2]$ neurons at each depth for both teacher and student. We also utilised batch normalisation for internal layers and relu non-activation. We generate 2500 samples from moons and circles dataset from scikit-learn[1] with noise=0.125. The teacher is trained on a classification task and the student is trained via. unsupervised distillation.

**Image Segmentation**: We trained FCN on top of the frozen backbones using the Pytorch implementation[2]. We trained a FCN32 head, for 200 epochs with a learning rate of $1e - 4$ and batch size of 20. The learning rate wa halved every 25 epochs. **CIFAR-100:** For a fair evaluation, we scaled the $\beta$ parameter of the adapted supervised distillation methods by a factor of 0.5, 1.0, 2.0, 4.0, 8.0, 16.0. We report below the optimal values we identified for each of the baseline methods.

## A.2  ABLATION STUDY

### A.2.1  IMPACT OF LOSS COMPONENTS

We perform this study to understand the importance of feature similarity and space similarity in our distillation framework. We provide results on CIFAR-100. In table 10, we report the unsupervised distillation performances from individual loss components. $L_{co}$ corresponds to the training with

---

[1]scikit-learn
[2]https://github.com/pochih/FCN-pytorch

Table 8: **Variation in performance for different** $\lambda$. We report top-1 accuracy

| Student/Teacher | $\lambda = 0$ | $\lambda = 0.25$ | $\lambda = 0.5$ | $\lambda = 1$ |
|---|---|---|---|---|
| ResNet8x4/32x4 | 72.05 | 73.44 | 73.90 | **73.91** |
| ResNet20/56 | 70.58 | **71. 34** | 71.11 | 70.68 |

Table 9: **Variation in performance for different nearest neighbourhood size**

| Methods | KNN-1 | KNN-51 | KNN-101 |
|---|---|---|---|
| SEED | 44.80 | 49.95 | 49.05 |
| BINGO | 48.90 | 53.92 | 53.19 |
| DisCo | 47.08 | 52.01 | 50.96 |
| CoSS (ours) | **49.41** | **54.68** | **53.70** |

Table 11: **Values for the loss scaling**

| Method | $\beta$ |
|---|---|
| ATT | 1000 |
| SP | 3000 |
| VID | 2.0 |
| RKD | 1.0 |
| PKT | 30000 |
| Factor | 200 |
| NST | 100 |
| CRD | 1.0 |
| SRRL | 1.0 |
| SKD | 100.0 |
| CoSS | 140 |

Table 10: **Unsupervised distillation with individual CoSS components**.

| Models | $L_{co}$ | $L_{ss}$ | $L_{coss}$ |
|---|---|---|---|
| Resnet8x4/32x4 | 72.05 | 73.53 | **73.90** |
| ResNet20/56 | 70.58 | 70.42 | **71.11** |

only feature similarity. This training is similar to BYOL(Grill et al., 2020), a SSL method. $L_{ss}$ corresponds to the results obtained by only relying on the space similarity loss. We observe that combining both both the losses performs better and thus is valuable for the process of distillation.

### A.2.2 VALUES OF $\lambda$

In table, we report the performance of CoSS for different values of $\lambda$. We perform this ablation on the CIFAR-100 dataset following the CRD training protocol. We observe that different architectures respond to different strength of the space-similrity component. Though other values of $\lambda$ provide better results, we selected $\lambda = 0.5$ in our experiments.

For ImageNet, using only $\lambda = 0$ degrades the performance by $0.8\%$ and $0.9\%$ for ResNet-18 and Efficientnet-b0 respectively.

### A.2.3 K-NEAREST NEIGHBOURS

In table 11, we report the KNN performance for different values of the nearest neighbourhoods. For the ImageNet distilled ResNet-18, we report KNN performance for sizes 1, 51 and 101. CoSS maintains better performance for a wide range of neighbourhood sizes.

### A.3 GOODNESS OF SPACE ALIGNMENT

To ascertain tthe quality of learnt embedding space after training, we compute the intersection-over-union of the nearest neighbours for both the student and the teacher. Formally, we report $iou_k = \frac{1}{|\mathcal{D}^{test}|} \sum_{x \in \mathcal{D}^{test}} \frac{K(f_s, x, \mathcal{D}^{train}, k) \bigcap K(f_t, x, \mathcal{D}^{train}, k)}{K(f_s, x, \mathcal{D}^{train}, k) \bigcup K(f_t, x, \mathcal{D}^{train}, k)}$. Where, $K$ returns a set of $k$ nearest samples to $x$ in the training data for the model $f_{(.)}$.

In table 12, we report the mean intersection-over-union of the neighbourhood for each input sample. It illustrates that student when trained with CoSS (our method) is able to learn the mapping which preserves the local neighbourhood in the latent space better than its peers. Also, as the neighbourhood size is increased we observe the trend of diminishing differences between the iou between different approaches. This indicates that baselines are only able to capture the local relationship at a much coarser level.

Table 12: **Comparison of common nearest neighbours for a student Resnet-18 with its teacher**. We report the Intersection-over-Union of the local neighbourhood for the ImageNet validation set. For each sample of the validation set, the nearest neighbours are searched in the training set. We observe that CoSS is better at modelling the local neighbourhood of its teacher than the other methods which rely on feature queues

| Methods | $IoU_1$ | $IoU_5$ | $IoU_{11}$ | $IoU_{21}$ |
|---------|---------|---------|------------|------------|
| SEED | 0.286 | 0.364 | 0.405 | 0.439 |
| BINGO | 0.289 | 0.360 | 0.397 | 0.427 |
| DisCo | 0.316 | 0.388 | 0.424 | **0.454** |
| CoSS (ours) | **0.338** | **0.399** | **0.430** | **0.454** |

## B   BATCH NORMALISATION FOR SPACE & SIMILARITY

Batch Normalisation (BN) was proposed to reduce the covariate shift which occurs during the mapping of inputs from one layer to another (Ioffe & Szegedy, 2015). Since its introduction, BN has found place in numerous deep learning architectures(He et al., 2016a;b; Xie et al., 2016). Here, we show how one can directly aim the distillation process to match student and network's embeddings and subsequently compare its performance with our formulation.

BN operates on a batch of input data, $X \in R^{b \times d}$ where batch size is $b$ and $d$ is the feature dimension. It first performs standardisation $\hat{X}_{:,i} = \frac{X_{:,i} - \mu_i}{\sigma_i}$ where, : denotes all the entries in the batch dimension and $\mu_i$, $\sigma_i$ are the mean and variances respectively for the $i^{th}$ feature dimension. The normalized values are then scaled by trainable parameters $\gamma_i$ and $\beta_i$ as:

$$Z_{:,i} = \gamma \hat{X}_{:,i} + \beta \qquad (4)$$

here, $\gamma_i$ and $\beta_i$ can be interpreted as affine transformations which operate independently for different spatial dimensions. We can utilise it to map the standardised student embeddings to the teacher's unnormalized embedding space. The corresponding loss can be defined as follows:

$$\mathcal{L}_{coss} = \frac{1}{b} \sum_{i=0}^{i<b} \mathcal{D}(Z_i^s, X_i^t)$$

where, $X_i^t$ is the teacher's embedding for the $i^{th}$ sample and $Z_i^s$ corresponds to the BatchNormalized student's embeddings. In table 13, we report the results using this approach on CIFAR-100 distillation task. We utilised mean-squared-error for the metric $D$.

Table 13: **CIFAR-100 unsupervised distillation with BN.**

| Methods | VGG13 | Resnet32x/8x | WRN-40/16 |
|---------|-------|--------------|-----------|
| BN | 74.01 | 72.22 | 73.42 |
| CoSS | 74.58 | 73.90 | 74.65 |

## C   TRANSFER LEARNING

The transfer learning experiment performed in the main paper consisted of fine-tuning a classification layer after extensive hyper-parameter tuning. In this section, we report the KNN10 accuracies obtained by the distilled ResNet-18s and Efficientnet-b0s. This way, we avoid the variability introduced by the final classification layer and can gauge the quality of embedding spaces of different networks.

From table 14 it can be observed that CoSS provides significant improvements over the baselines. This trend is similar to the top-1 results reported in the main paper.

Table 14: **Transfer learning evaluation of distilled ResNet-18 and Efficientnet-b0.** We report the KNN-10 accuracy

| Method | CIFAR-10(Krizhevsky, 2009) | | CIFAR-100 | | STL-10(Coates et al., 2011) | | Caltech-101(Fei-Fei et al., 2006) | | Pets(Parkhi et al., 2012) | | Flowers(Nilsback & Zisserman, 2006) | | DTD(Cimpoi et al., 2014) | |
|---|---|---|---|---|---|---|---|---|---|---|---|---|---|---|
| | ResNet-18 | Eff-b0 | ResNet-18 | Eff-b0 | ResNet-18 | Eff-b0 | ResNet-18 | Eff-b0 | ResNet-18 | Eff-b0 | ResNet-18 | Eff-b0 | ResNet-18 | Eff-b0 |
| SEED | 80.86 | 84.96 | 52.09 | 60.38 | 91.20 | 92.97 | 73.60 | 76.78 | 60.40 | 66.53 | 47.18 | 65.54 | 56.91 | 61.38 |
| BINGO | 84.01 | 85.61 | 58.09 | 60.43 | 93.31 | 92.71 | 77.02 | 76.92 | 66.50 | 67.76 | 58.79 | 68.78 | 60.00 | 61.97 |
| DisCo | 84.45 | 86.80 | 56.80 | 62.68 | 92.46 | 92.00 | 77.05 | 76.58 | 64.95 | 65.69 | 59.85 | 64.94 | 61.28 | 62.18 |
| CoSS (ours) | **85.92** | **87.61** | **60.14** | **66.06** | **92.33** | **93.56** | **77.13** | **79.19** | **67.27** | **68.79** | **64.87** | **72.82** | **62.39** | **64.26** |

## D ROBUSTNESS

In table 15, we provide the results of KNN10 evaluation of the Efficient-b0 on the robustness benchmarks. Similar to the ResNet-18, Efficientnet-b0 distilled using CoSS is significantly more robust.

Table 15: **Robustness evaluation of ImageNet distilled Efficientnet-b0.** To emphasize on the quality of embeddings learnt, we report the KNN-10 accuracy. We highlight the best performance in **bold**

| Method | ImageNet-v2(Recht et al., 2019) | | | ImageNet-S(Wang et al., 2019) | ImageNet-C(Hendrycks & Dietterich, 2019) | | | | | | | | | | | | | | |
|---|---|---|---|---|---|---|---|---|---|---|---|---|---|---|---|---|---|---|---|
| | MF | Tr | Top | | brightness | contrast | defocus | elastic | fog | frost | gaussian | glass | impulse | jpeg | motion | pixelate | shot | snow | zoom |
| SEED | 40.60 | 48.17 | 53.72 | 12.72 | 50.58 | 45.69 | 35.52 | 46.82 | 36.48 | 39.81 | 37.32 | 34.15 | 26.41 | 39.78 | 39.60 | 45.04 | 36.14 | 33.49 | 27.54 |
| BINGO | 41.47 | 49.15 | 54.65 | 13.15 | 51.45 | 46.62 | 35.83 | 47.47 | 38.29 | 40.73 | 37.69 | 34.73 | 27.30 | 40.51 | 40.22 | 44.31 | 36.37 | 33.84 | 26.81 |
| DisCo | 41.10 | 48.49 | 54.32 | 11.17 | 52.05 | 47.22 | 37.55 | 47.42 | 33.94 | 37.71 | 37.09 | 33.82 | 24.26 | 41.55 | 38.54 | 46.61 | 35.34 | 29.22 | 23.37 |
| CoSS (ours) | **44.40** | **52.07** | **58.13** | **13.50** | **55.68** | **51.71** | **42.04** | **51.90** | **41.33** | **43.66** | **41.49** | **39.59** | **28.62** | **45.77** | **43.74** | **50.87** | **40.30** | **36.69** | **32.19** |

## E VISUALISATIONS

In figure 5, we provide the visualisations of the self-attention maps generated by the CoSS student on randomly selected images from the MSRA10K dataset. It is interesting to note that each head focuses on different aspects of an input image such as the foreground, background, or discriminating features.

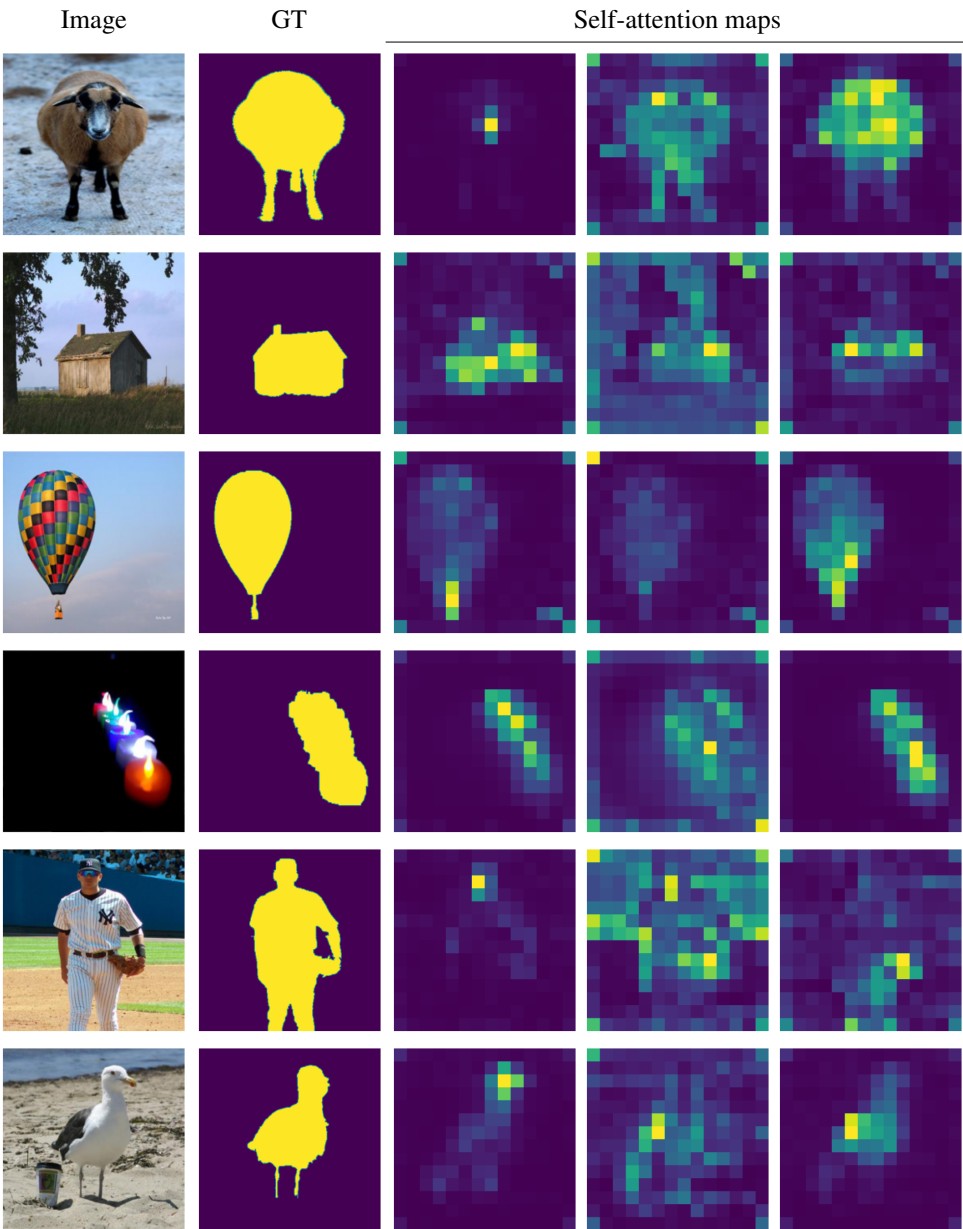

Figure 5: **Self-attention maps of [CLS] token from the** 3 **ViT-tiny heads.** The bright regions denote the patches of interest to the model

