# OpenReview forum: "Efficient Unsupervised Knowledge Distillation with Space Similarity"
_ICLR.cc/2024/Conference — Submitted to ICLR 2024_

### Official Review · Reviewer_u1Un · 2023-10-17

**Soundness:** 2 fair
**Presentation:** 2 fair
**Contribution:** 2 fair
**Rating:** 5
**Confidence:** 4

**Summary:**

In this study, the authors discuss boosting performance of knowledge distillation without use of ground-truth labels. Knowledge distillation usually uses combinations of human annotations and features/embeddings extracted from teacher models, but this study discusses knowledge distillation in an unsupervised setting and introduces cosine similarity and space similarity (CoSS) training objective to help student models learn to mimic teacher models' embedding structure. The proposed approach is numerically assessed mainly for ResNet-18 and EfficientNet-b0 in image classification tasks and image/video instance segmentation tasks. The evaluation also involves GPU memory requirement per method, and the result seems comparable to that of SEED baseline. Based on those results, the authors confirm the improvements by the proposed approach in many scenarios (models x methods x tasks) in efficient manners.

**Strengths:**

Given that majority of the existing studies on knowledge distillation requires human annotations to improve model performance, the reviewer sees some originality of this study. There are not many existing studies that discuss knowledge distillation in a pure unsupervised learning setting for target tasks.

The strongest point of this paper may be a lot of experiments the authors conducted such as multiple image classification tasks and image/video object segmentation tasks. Even though each of the experiments is not well described (e.g., lacking justification of baselines and hyperparameter tuning for those methods), the reviewer wants to recognize the attempt to show how the proposed method generalizes.

This study provides not only quantitative assessments but qualitative assessments e.g., Figs. 3 - 5. While Figs. 4 and 5 seem not much inconclusive, specifically with respect to BINGO+, the provided examples helped the reviewer see the representations learned with CoSS is more similar to that of teacher than SEED.

**Weaknesses:**

Even though the reviewer appreciates many experiments for various tasks, this paper lacks descriptions and justifications of the baselines and how tuned the baseline methods are.

The reviewer also believes that it is unfair to use supervised methods without supervised signals (human annotations) as baselines unless their hyperparameters are tuned without the annotations, but neither hyperparameter tuning nor choices is described in this paper.

For the same reason, the reviewer also has a concern about reproducibility of this work. Appendix A.1 is not detailed enough (how teacher model is trained, architectures of additional trainable layers, baseline hyperparameters, etc) and does not cover all the experiments (at least) in the main body.


This paper also lacks clarity and needs improvement in writing.
- The reviewer needs more clarifications in the description of the additional evaluation with kNN as it's not convincing. While Section 5.1 explains it is because the approach allows them to evaluate the methods directly without the need of tuning parameters, it is still dependent on the choice of $k$, which is not justified but heuristic.
- In Section 1, the reduction of GPU memory requirement is emphasized, but the GPU memory requirement is not defined in this paper. How was it measured? If the teacher's embeddings are pre-computed and cached, it maybe easily save GPU memory usage as much as the proposed method does.
- There are many typos and grammatical errors:
  - "distilled student" should be replaced with "trained student", as (knowledge of) teachers are distilled into students, and students are not distilled
  - and Attention transfer -> and attention transfer
  - "(i) feature queues (ii) contrastive objectives (iii) heavy augmentations (iv) and custom batch composition" -> "(i) feature queues, (ii) contrastive objectives, (iii) heavy augmentations, and (iv) custom batch composition"
  - Some notations are not defined where used. e.g., $\widehat{A_t^T}$ and $\widehat{A_s^T}$ in Eq. (4) $\lambda$ in Eq. (5)
  - DisCo vs. DISCO
  - section vs. Section
  - Table vs table
  - Figure vs. figure
  - In the Appendix -> In Appendix
  - "We compute nearest neighbour in the size 10 neighbourhood of the sample" -> "We choose 10 nearest neighbours from the training samples"
  - Ericsson et. al. Ericsson et al. (2021) -> Ericsson et al. (2021)
  - ImageNet vs. imagenet vs. Imagenet
  - "two settings 1. traditional ... 2. Following" -> "two settings. 1) Traditional ... 2) Following"
  - "as reported in 4" -> "as reported in Table 4"
  - We'd like -> We would like (not an error, but suggested)

**Questions:**

Questions
- How did the authors reduce embedding space to make plots in Fig. 3 (c) and (d)?
- What is the difference between $\hat{A}_t$ in Eq. (3) and $\widehat{A_t^T}$ in Eq. (4)? (same for $\hat{A}_s$ vs. $\widehat{A_s^T}$)
- Why is SSKD (Xu et al., 2020) as referred as part of unsupervised distillation (Section 2.2)? SSKD does use human annotations.

Suggestions

Human-annotation-free knowledge distillation is also discussed in the NLP community, and the authors may want to discuss the proposed
 or similar approach for NLP tasks in future work. For instance,
- Embeddings-based KD: Reimers and Gurevych (2020) ["Making Monolingual Sentence Embeddings Multilingual using Knowledge Distillation"](https://aclanthology.org/2020.emnlp-main.365/)
- Logits-based KD: Gupta et al. (2023) ["Cross-Lingual Knowledge Distillation for Answer Sentence Selection in Low-Resource Languages"](https://aclanthology.org/2023.findings-acl.885/)

---

> ### Author Response · Authors · 2023-11-11
> **Author Response (1/2)**
>
> **We express our gratitude to the reviewer for investing their valuable time in reviewing our work. Their insightful observations have been duly noted, and we address them sequentially below.**
>
> > Even though the reviewer appreciates many experiments for various tasks, this paper lacks descriptions and justifications of the baselines and how tuned the baseline methods are.
>
> The baselines provided in the paper are introduced and discussed in the related section. Regarding the tuning, as highlighted in A.1, we utilise their official implementations for reproducing results (along with official hyper-parameters). SEED, DisCo, BINGO all use identical settings for training and thus, by following SEED, we are able to make a fair comparison with other methods as well. More importantly, we follow SEED’s training hyper-parameters such as augmentations, epoch, batch-size, optimiser etc for our work thus, making the results in sec 5.1 meaningful and valid. For all the subsequent experiments 5.3 and onwards, we employ the backbone obtained from sec 5.1. For sec 5.2, the hyper-param sweep is performed for all models (from 5.1) for all datasets following Ericsson et al. If there are any more specific details that the reviewer would like to be reported, we would be happy to incorporate it.
>
> > The reviewer also believes that it is unfair to use supervised methods without supervised signals (human annotations) as baselines unless their hyperparameters are tuned without the annotations, but neither hyperparameter tuning nor choices is described in this paper.
>
> If the reviewer is referring to supervised methods in experiment 5.7, we would like to apologise for the oversight. As mentioned in the paper, we followed the official implementation of CRD for generating these results. For a fair comparison, for each baseline, we scaled the recommended loss multiplier $\beta$ by factors of **{0.5, 1, 2, 4, 8 ,16}** and selected the value which yielded the highest top-1 for the **Resnet56/20** teacher-student pair. Note that the contribution of supervision and logics based KD is set to 0. Keeping this $\beta$ fixed, we then performed distillation on all the other teacher-student pairs and reported the results. For CoSS, we performed the hyper-param search for **Resnet56/20** and fixed the obtained $\beta=140$  for all other teacher-student pair as well. We will be adding the above mentioned information along with shortlisted $\beta$ for each baseline to A.1 in the newer version. If there is any other additional that the reviewer would like us to cover, we will happily oblige.
>
> > For the same reason, the reviewer also has a concern about reproducibility of this work. Appendix A.1 is not detailed enough (how teacher model is trained, architectures of additional trainable layers, baseline hyperparameters, etc) and does not cover all the experiments (at least) in the main body.
>
> We would like to clarify the reproducibility aspect of our work. As mentioned in A.1, we have adopted SEED’s training hyper-parameters (teacher, scheduling, epochs, optimizer, augmentations etc) for the ImageNet distillation experiments. The cited paper thoroughly covers the necessary information. However, we would like to apologise for omitting the overall loss scaling value which was learnt from the CIFAR experiments as mentioned in the previous response. We directly apply the scaling factor ($\beta$=140) learnt from CIFAR to the ImageNet experiments as well. Moreover, post decision deadline, we will be sharing the official code for our work.
>
> > The reviewer needs more clarifications in the description of the additional evaluation with kNN as it's not convincing. While Section 5.1 explains it is because the approach allows them to evaluate the methods directly without the need of tuning parameters, it is still dependent on the choice of k, which is not justified but heuristic.
>
> Thank you for the insightful suggestion. kNN performance indeed depends on the choice of the neighbourhood. The choice of $k$, as reported in Section 5.1, is kept to be in-line with the evaluation setting adopted in SEED. The authors of SEED chose the value of $k=10$ in their evaluation. Though this justifies as to why we have used $k=10$, it does not explain as to why only $k=10$ should be preferred. To circumvent this issue, we have reported kNN with k={1,51,101} in A.2.2, where we see a similar trend across different k.

---

> ### Author Response · Authors · 2023-11-11
> **Author Response (2/2)**
>
> > In Section 1, the reduction of GPU memory requirement is emphasised, but the GPU memory requirement is not defined in this paper. How was it measured? If the teacher's embeddings are pre-computed and cached, it maybe easily save GPU memory usage as much as the proposed method does.
>
> Thank you for pointing this out. We measured GPU memory as the maximum GPU space occupied by the method while processing an input batch. We reported the mean computed over the first 100 training batches. Training batch size for all methods is the same (as per implementation notes). Caching the representations will only allow us to not load the teacher model onto the GPU for computing embeddings. However, there are few downsides with this.
>
>  (1) It will only save the GPU memory required by the teacher, since the teacher embeddings are required to be on GPU for loss (gradient) computation and hence does not free up GPU space required for embedding themselves.
>
> (2) Teacher Resnet-50 utilises considerably less GPU space (~100MB) when loaded only for inference/embedding generation. Additionally, the cost of a forward pass for this computation is negligible compared to the overall time some methods take.
>
> (3) Each method (including CoSS) loads the teacher onto the GPU, hence if an optimisation can help other methods, it should also help CoSS.
>
> (4) Usage of heavy augmentations like CutMix as in BINGO, makes it hard to pre-process and store all the teacher embeddings and hence requires an online computation of embeddings.
>
> We will be adding these omitted details to the appendix.
>
> > How did the authors reduce embedding space to make plots in Fig. 3 (c) and (d)?
>
> Thank you for highlighting this aspect. The MLP architecture selected for this task performs the mapping 2->4->8->4->2->2. The 2D input is mapped via hidden layers to the 2D embedding layer which subsequently feeds into the 2D classification layer. We have visualised the 2D embedding space as learnt by the teacher and student models coming from the penultimate layer in Fig 3. (b,c,d). We will be adding this information along with more details to reproduce the plots in the appendix.
>
> > What is the difference between $\hat{A}_t$ and $\widehat{A^T_t}$ in Eq. (3) and in Eq. (4)?
>
> We have used $\hat{.}$ to denote the l2 normalisation of a matrix along the last dimension (columns). This means that $\hat{A}_t$ is the batch of L2 normalised teacher’s embeddings stacked along rows. $\widehat{A^T_t}$ on the other hand first applies the matrix-transpose operation and then performs the l2 normalisation. Effectively, what we obtain is, l2 normalization along the space direction.
>
> > Why is SSKD (Xu et al., 2020) referred to as part of unsupervised distillation (Section 2.2)? SSKD does use human annotations.
>
> We mentioned SSKD along with CompRes and SEED to highlight a high degree of similarity in the KD objectives of these methods though they are for supervised and unsupervised tasks respectively. We can now understand the confusion our phrasing might create, and duly apologise for it. To rectify this, we have added a short intro to SSKD in the supervised section and reiterated on the fact that SSKD is supervised when mentioned in the unsupervised section.
>
> $\textbf{Typos}$: We once again thank the reviewer for providing extremely thorough feedback on our work. We have addressed and fixed the grammatical errors and typos in our newer draft. We will be uploading it once we have incorporated suggestions from other reviewers as well.
>
> > Human-annotation-free knowledge distillation is also discussed in the NLP community, and the authors may want to discuss the proposed or similar approach for NLP tasks in future work
>
> The reviewer has provided a noteworthy perspective. We will be adding a discussion to highlight, compare and contrast unsupervised distillation methods across domains.

---

> > ### Comment · Reviewer_u1Un · 2023-11-18
> >
> > The reviewer thanks the authors for their clarifications. Some of the reviewer's concerns were addressed.
> >
> > > We measured GPU memory as the maximum GPU space occupied by the method while processing an input batch. We reported the mean computed over the first 100 training batches. Training batch size for all methods is the same (as per implementation notes).
> >
> > This is still a concern for the reviewer. Taking an average of peaked GPU spaces over batches should be noisy as only one data point is extracted at each batch, and the reviewer suggests discussing the GPU requirements using a more practical metric such as (how much GPU memory is consumed by a method) * (how long the method takes to complete the training)
> >
> > Also, the reviewer highly recommends using different notations for multiple A hats. Those seem like very confusing.

---

> > > ### Author Response · Authors · 2023-11-19
> > > **Author Response**
> > >
> > > > **This is still a concern for the reviewer. Taking an average of peaked GPU spaces over batches should be noisy as only one data point is extracted at each batch, and the reviewer suggests discussing the GPU requirements using a more practical metric such as (how much GPU memory is consumed by a method) * (how long the method takes to complete the training)**
> > >
> > > We thank the reviewer for their continued engagement. We’d like to highlight that the GPU space occupied is computed at each iteration and peak memory is often used as a reliable metric[1]. Though it may be noisy, by reporting the mean over the first 100 iterations we reduce its impact. We’d also like to thank the reviewer for the suggested metric. We will be adding it to our results under Total-Training-Resources (TTR). Moreover, based on [1], we will also be providing the trend of (GPU, time) resources consumed per iteration over the course of training.
> > >
> > > > **Also, the reviewer highly recommends using different notations for multiple A hats. Those seem like very confusing.**
> > >
> > > We thank the reviewer for their  insightful feedback and valuable recommendations regarding the notations. We have polished our notations to reduce confusion. We will be uploading the new version in the next few hours incorporating requested changes and results.
> > >
> > > **References**
> > > [1] Cui, J., Wang, R., Si, S., & Hsieh, C. J. (2023, July). Scaling up dataset distillation to imagenet-1k with constant memory. In International Conference on Machine Learning (pp. 6565-6590). PMLR.

---

> > > > ### Comment · Reviewer_u1Un · 2023-11-20
> > > >
> > > > > We thank the reviewer for their continued engagement. We’d like to highlight that the GPU space occupied is computed at each iteration and peak memory is often used as a reliable metric[1]. Though it may be noisy, by reporting the mean over the first 100 iterations we reduce its impact. We’d also like to thank the reviewer for the suggested metric. We will be adding it to our results under Total-Training-Resources (TTR). Moreover, based on [1], we will also be providing the trend of (GPU, time) resources consumed per iteration over the course of training.
> > > >
> > > > > References [1] Cui, J., Wang, R., Si, S., & Hsieh, C. J. (2023, July). Scaling up dataset distillation to imagenet-1k with constant memory. In International Conference on Machine Learning (pp. 6565-6590). PMLR.
> > > >
> > > > While the authors claim that the peak memory is often used as a reliable metric in [1], the reviewer couldn't shortly find any such claims in the paper. Where can we find such claims and data points in [1]?
> > > >
> > > > The reviewer is still concerned about the resource usage assessment since it uses only the first 100 iterations (not 100 epochs), which is largely affected by the batch size. Thus, the reviewer suggests removing GPU (GB) columns.
> > > > Besides those, the reviewer wants to know how the runtime in this study was measured. Is it per sample? or per batch?

---

> > > > > ### Author Response · Authors · 2023-11-20
> > > > > **Author Response**
> > > > >
> > > > > > **While the authors claim that the peak memory is often used as a reliable metric in [1], the reviewer couldn't shortly find any such claims in the paper. Where can we find such claims and data points in [1]?**
> > > > >
> > > > > In the main paper, they don't cover what their metric 'GPU memory' corresponds to, but, in the appendix Tables 4,5 they specify the usage of peak GPU consumption as the metric. These numerical results correspond to the Figures 2,3 in their main paper (their section 5.3 highlights this fact) . Based on this, we believe that throughout their paper, the memory they refer to is indeed peak GPU memory consumed.
> > > > >
> > > > > > **The reviewer is still concerned about the resource usage assessment since it uses only the first 100 iterations (not 100 epochs), which is largely affected by the batch size. Thus, the reviewer suggests removing GPU (GB) columns.**
> > > > >
> > > > > We would like to highlight that we extended iterations from 100 to 1000. Each iteration corresponds to the forward-backward pass over a constant batch size of $256$ for all methods. Therefore ,we collect 1000 data points over which we report the mean values of runtime and GPU consumption. We found that the variance between datapoints is less than 1-10 MBs.
> > > > >
> > > > > The GPU consumption indeed depends on the batch size of the approaches, however, we have considered identical batch size for all method. We will be providing results over multiple batch configurations (64, 128) to demonstrate how the GPU requirement varies. Our initial estimate is that Memory requirement (absolute values) should decrease for all methods, but it should not impact the relative ordering of the approaches.
> > > > >
> > > > > In our view, peak-memory consumption is an important metric because it reflects the differences in resource requirements for identical training settings. We believe that providing ablation over batch size should strengthen our message. Moreover, TTR, as kindly recommended by the reviewer, encapsulates the complete training dynamics (epochs, batch sizes, iterations) which is batch independent. It is computed as peak-gpu-memory of a batch * number of iterations * number of epochs and by varying the batch size, the corresponding number of iterations inversely varies.
> > > > >
> > > > > > **Besides those, the reviewer wants to know how the runtime in this study was measured. Is it per sample? or per batch?**
> > > > >
> > > > > The runtime is measured per batch (batch size=256) and **does not include** the data loading time. We report the mean runtime gathered from 1000 iterations (i.e. 1000 datapoints).

---

> > > > > ### Author Response · Authors · 2023-11-20
> > > > > **Peak memory ablation with Batch**
> > > > >
> > > > > Below we report peak memory usage with varying batch sizes. We observe that the compute requirement varies (approx) linearly with the batch size. Under similar batch budget, our approach requires 3x less compute than BINGO.
> > > > >
> > > > > | Resnet-18 | BINGO | DisCo | SMD   | SEED | CoSS     |
> > > > > |-----------|:-----:|:-----:|-------|------|----------|
> > > > > | b=64      | 5.57  | 4.21  | 3.16  | 2.41 | **1.77** |
> > > > > | b=128     | 10.74 | 7.78  | 6.03  | 4.53 | **3.22** |
> > > > > | b=256     | 21.10 | 14.89 | 11.74 | 8.77 | **6.10** |

---

> > > > > > ### Comment · Reviewer_u1Un · 2023-11-22
> > > > > >
> > > > > > The reviewer thanks for the authors' follow-up comments.
> > > > > >
> > > > > > > > We’d like to highlight that the GPU space occupied is computed at each iteration and peak memory is often used as a reliable metric[1].
> > > > > >
> > > > > > > In the main paper, they don't cover what their metric 'GPU memory' corresponds to, but, in the appendix Tables 4,5 they specify the usage of peak GPU consumption as the metric. These numerical results correspond to the Figures 2,3 in their main paper (their section 5.3 highlights this fact) . Based on this, we believe that throughout their paper, the memory they refer to is indeed peak GPU memory consumed.
> > > > > >
> > > > > > It does not support the initial statement "peak memory is often used as a reliable metric[1]", but sounds more like the authors' opinion.
> > > > > >
> > > > > >
> > > > > > It is good to learn that the authors use the same batch size across methods. The reviewer suggests adding the clarifications to the paper.
> > > > > >
> > > > > > Lastly, the reviewer wonders why this work does not have Hinton et al.'s standard KD (but without the cross entropy term), which will be a strong baseline in terms of TTR (and peak GPU memory though the reviewer feels the metric is no longer necessary because of TTR)

---

> ### Author Response · Authors · 2023-11-22
> **Author Response**
>
> We express our gratitude to the reviewer for their consistent and valuable feedback.
>
> > **It does not support the initial statement "peak memory is often used as a reliable metric[1]", but sounds more like the authors' opinion.**
>
> We appreciate the reviewer feedback and subsequently have removed the peak-memory usage from Table 2.
>
>
> > **It is good to learn that the authors use the same batch size across methods. The reviewer suggests adding the clarifications to the paper.**
>
> We appreciate the reviewer's acknowledgement. We will incorporate the suggested clarifications into the paper.
>
> > **Lastly, the reviewer wonders why this work does not have Hinton et al.'s standard KD (but without the cross entropy term), which will be a strong baseline in terms of TTR**
>
> We followed prior works (SEED, BINGO etc.) in the domain of unsupervised distillation for setting up our baselines.
>
> [**Edited**] We agree that standard KD [1] (without CE) would be a strong baseline in terms of TTR, but, we also suggest that this baseline may not be a valid baseline for knowledge distillation. Standard KD formulates the distillation process as KL(Q||P), where $Q$ and $P$ corresponds to the teacher and student probabilities over the output space respectively. Without the CE term, it would be $\sum_{x \in \mathcal{X}} Q(x) log(Q(x))$ which is independent of the student. Also, with the alternative formulation which many implementations follow (such as CRD[2]), we have $KL(P||Q)$ and in this case we have a training objective independent of the teacher.
>
> Since, the student will not learn anything from the teacher, we suggest that adding a baseline which diverges from the core methodology of knowledge distillation may not add value.
>
> [1] Hinton, Geoffrey, Oriol Vinyals, and Jeff Dean. "Distilling the knowledge in a neural network." arXiv preprint arXiv:1503.02531 (2015).
>
> [2] Tian, Yonglong, Dilip Krishnan, and Phillip Isola. "Contrastive representation distillation." ICLR (2020).

---

> > ### Comment · Reviewer_u1Un · 2023-11-22
> >
> > The reviewer is afraid that there are some misunderstandings regarding Hinton et al.'s KD.
> > Its loss function is a linear combination of cross entropy and KL divergence terms, called soft and hard targets respectively.
> > Without the cross entropy term, it will be only KL divergence between softened class probabilities from teacher and student models. Thus, the student can learn from the teacher. In fact, one of the papers the reviewer suggested above shows that student models learned from teacher models without cross entropy term.
> >
> > > Logits-based KD: Gupta et al. (2023) ["Cross-Lingual Knowledge Distillation for Answer Sentence Selection in Low-Resource Languages"](https://aclanthology.org/2023.findings-acl.885/)
> >
> > The reviewer believes this is the formulation defined Hinton et al.'s KD paper (using a relative weight for the linear combination), and even early KD studies such as FitNets (2015) https://arxiv.org/abs/1412.6550 use the formulation while CRD paper was published in 2020.
> >
> > For these reasons, the reviewer disagrees with the authors on the statement that student models won't learn anything from teachers by minimizing a standard KD loss without cross entropy term. This also means that the reviewer still believes that it is a strong baseline in terms of 1) TTR since there are no additional layers/modules to use other than the original student and teacher models and 2) accuracy when the temperature term is well-tuned.

---

> > > ### Author Response · Authors · 2023-11-23
> > > **Thank you!**
> > >
> > > **As the discussion period draws to a close, we extend our sincere appreciation to the reviewer for their active engagement and constructive criticism throughout the rebuttal phase. We cordially invite the reviewer to evaluate the revised manuscript, now updated to encompass the recommended changes based on their valuable insights.**

---

> ### Author Response · Authors · 2023-11-22
> **Author Response**
>
> We thank the reviewer for providing clarification.
>
> To answer reviewer's query with the standard KD loss (with only KL), we are sharing results below for the ImageNet (ResNet-18, EfficientNet-b0), CIFAR-100 tasks with the aforementioned loss. Moreover, we also share the runtime and TTR for the ResNet-18 distillation on ImageNet.
>
> **ImageNet**
>
> | **Method** | **ResNet-18 (top-1)** | **ResNet-18 (kNN)** | **Eff-b0 (top-1)**   | **Eff-b0 (kNN)**  |
> |--------|-------------|----------------|-----------|-----------|
> | KD |   58.16   |   54.31 |   61.49  |  56.10  |
> | CoSS   |  **59.24**  | **55.04**  |  **63.55**  | **58.32**   |
>
> **CIFAR-100**
>
> | **Method** | **Resnet20/56** | **Resnet8x4/32x4** | **VGG8/13**   | **WRN16/40**  |
> |--------|-------------|----------------|-----------|-----------|
> | KD |    68.69   |   72.52    |  73.33    | 73.01     |
> | CoSS   | **71.11**   | **73.90**      | **74.58** | **74.65**     |
>
> **Runtime**
>
> | **Method** | **Runtime** | **TTR** |
> |--------|-------------|---|
> | KD |  0.19  |   322.26 |
> | CoSS   | 0.19 | 322.26 |
>
> The difference in runtime was in micro-scale which was ironed out due to rounding off. CoSS and KD utilise the least amount of TTR compared to others.
>
> The temperatures we used are $\tau_{student}=0.1$ and $\tau_{teacher}=0.07$ following the SSL adaptation of standard KD by [1].
>
> We agree with reviewer's feedback regarding KD being a strong baseline hence, we will be adding these results to the paper.
>
> **References:**
>
>     [1] Caron, Mathilde, Hugo Touvron, Ishan Misra, Hervé Jégou, Julien Mairal, Piotr Bojanowski, and Armand Joulin. "Emerging properties in self-supervised vision transformers." In Proceedings of the IEEE/CVF international conference on computer vision, pp. 9650-9660. 2021.

---

### Official Review · Reviewer_sCnR · 2023-10-30

**Soundness:** 3 good
**Presentation:** 3 good
**Contribution:** 2 fair
**Rating:** 3
**Confidence:** 4

**Summary:**

The paper introduces an approach to unsupervised knowledge distillation that avoids reliance on a queue or contrastive loss. It identifies and addresses the non-homeomorphic issue in cosine similarity by enhancing both the Cosine similarity and Space Similarity between the student and teacher models. Compared to existing methods, this approach demonstrates significant improvement and reduces both training time and GPU memory usage.

**Strengths:**

1) The proposed approach reduces training time and memory usage for unsupervised knowledge distillation, addressing the non-homeomorphic problem in cosine similarity by adding Space Similarity.

2) Students trained by the proposed method have strong transferability and remain robust even in the face of input distribution shifts.

3) This paper is easy to understand and implement.

**Weaknesses:**

1) The contribution of the method is limited. A similar idea has already been discussed in [1]. For each pair of prediction vectors from the student and teacher within a batch $A^s, A^t \in \mathbb{R}^{b\times d}$, [1] proposed inter-relation loss $\frac{1}{b}\sum^b_{i=1}g(A^s_{i,:}, A^t_{i,:})$, intra-relation loss $\frac{1}{d}\sum^d_{j=1}g(A^s_{:,j}, A^t_{:,j})$ where $g(\cdot,\cdot)$ is a distance function, $b$ is batch size and $d$ is the feature dimensions. [1] employ those loss functions on the logits vectors where $d$ is the number of classes. According to Figure 2, the proposed method is similar to [1]. The difference is the proposed method employs those loss functions on the feature vectors.

2) The proposed method is behind some existing methods in the large dataset (ImageNet 1K) in Table 1. There is also an absence of a comparison with state-of-the-art methods, such as SMD[2].

[1] Huang, T., You, S., Wang, F., Qian, C., & Xu, C. (2022). Knowledge distillation from a stronger teacher. Advances in Neural Information Processing Systems, 35, 33716-33727.

[2] Liu, H., & Ye, M. (2022, October). Improving Self-supervised Lightweight Model Learning via Hard-Aware Metric Distillation. In European Conference on Computer Vision (pp. 295-311). Cham: Springer Nature Switzerland.

**Questions:**

3) I am confused about $L_{ss}$ and $L_{co}$. For the representations matrix $\hat{A_t}$, $\hat{A_s}$ $\in R^{b\times d}$. What is the dimensions of the $A_{I}$ and $A_{II}$ in the Equation 3 and Equation 4? As mentioned in the paper, $A^i_s$ is only compared with
$A^i_t$. If $L_{co}$ calculating the cosine similarity for each pair of features in the input batch, the dimensions of ${A_I}$ should be $R^{b}$

4) There is a lack of ablation study for $L_{ss}$, $L_{co}$. Is the Space Similarity sensitive to the batch size? Could the authors conduct experiments using various batch sizes? How does the performance of the proposed method compare to SMD?

5) Can authors compare the proposed method with the method mentioned in reference [1]?

6) Can the authors provide additional details about ViT training in Tables 1 and 6? Were all ViTs trained from scratch?

---

> ### Author Response · Authors · 2023-11-12
> **Author Response (1/2)**
>
> **We would like to express our sincere gratitude to the reviewer for their time to evaluate and provide feedback on our work. In particular, we would like to thank the reviewer for recognizing the simplicity and efficiency of our approach. We are also grateful for other suggestions. We address each of the noteworthy points they raised.**
>
> **Note: We will be uploading the updated version of the paper once we incorporate the changes proposed by all the reviewers. We thank the reviewer for their patience in this regard.**
>
> > **W1**: The contribution of the method is limited. A similar idea has already been discussed in [1]...
>
> Thank you for bringing this to our notice. Indeed, [1] has a similar objective where they consider intra-class responses for their supervised setting. One subtle yet important difference in implementations of the two is their treatment of intra-class responses prior to the similarity computation. In [1], **the authors normalise the features prior to computing their inter and intra class losses**. We do not perform this step so as to avoid the loss of original manifold information which a normalisation operation induces. Another important difference is in the motivation and subsequently the setting to which these two strategies are applied. [1] utilises the class-wise scores as means to capture (class) prior from the teacher which consequently is modelled as a correlation maximisation problem. For this purpose, it makes sense to pre-normalize the logits to probability over classes before computing the intra-class loss. On the other hand, we have focused on modelling the manifold space of a teacher for which the space-wise similarity allows us to rectify the loss of information caused by the normalised feature-wise similarity. For this, we do not normalise features prior to computing space similarity. Lastly, we hope to remind the reviewer that unsupervised vs supervised is also usually considered as a distinction. We have added this discussion with [1] in the related work of our new draft.
>
>
> > **W2.A**: The proposed method is behind some existing methods in the large dataset (ImageNet 1K) in Table 1.
>
> Thank you for noticing this, we agree that the proposed method is behind other methods in only terms of top-1 in Table 1. A gentle reminder is that we can achieve this with significantly less computing efforts. For example, BINGO which performs 0.7% better on (Resnet-18) top-1 requires roughly **9x more time** to train. And, for DisCO in case of efficientnet-b0, the GPU compute and training time required increases **~5x**. More importantly, additional experiments in 5.2 and onwards showed that the model from  the proposed method can still serve as an alternative backbone for various tasks where it often performs better than other models.
>
>  > **W2.B**: There is also an absence of a comparison with state-of-the-art methods, such as SMD[2]
>
> Thanks for the suggestion, we didn’t compare SMD because of its different settings. For example, BINGO, DisCo, and SEED are using the same evaluation protocol, while the one SMD uses is different.
>
> We are currently running distillation on ImageNet using SMD and will report the results in the coming days. Below are the preliminary results on CIFAR-100 for SMD compared with CoSS.
>
> | **Method** | **Resnet20/56** | **Resnet8x4/32x4** | **VGG8/13**   | **WRN16/40**  |
> |--------|-------------|----------------|-----------|-----------|
> | SMD[2]    | 70.33       | 71.41          | 74.10     | **74.93** |
> | Ours   | **71.11**   | **73.90**      | **74.58** | 74.65     |
>
> > **Q3**: I am confused about L_ss and L_co. For the representations …
>
> The dimension for $A_{I}$ and $A_{II}$ is $R^{b \times d}$ similar to those of $\hat{A}_t \in R^{b\times d}$ and $\hat{A}_s \in R^{b\times d}$. This is because $A_I = \hat{A}_t \odot \hat{A}_s$, here $\odot$ is the point operation performing element wise multiplication. During loss computation, we then perform summation along the columns which yields the dot product.
>
> > **Q4.A**: There is a lack of ablation study for L_ss, Lco. Is the Space Similarity sensitive to the batch size?
>
> Thank you for this suggestion. Below we report the performance of CoSS on different input batch sizes analogous to Table 7 in appendix A.2. As we can note, the performance of CoSS remains performant at small batch sizes well. For larger batches, there is a slight drop in performance.
>
>
> | **Method** | $b=32$ | $b=64$ | $b=128$   | $b=256$  | $b=512$ | $b=1024$ |
> |--------|-------------|----------------|-----------|-----------|-----------|-----------|
> | Resnet8x4/32x4   |   73.21    |  73.62        |  73.90     | 73.35     | 72.57 |  72.09  |
> | Resnet20/56   |   70.34    |  71.32        |  71.11     | 71.05     | 70.59 | 70.11

---

> ### Author Response · Authors · 2023-11-12
> **Author Response (2/2)**
>
> > **Q4.B**: How does the performance of the proposed method compare to SMD?
>
> The experiments are underway and we will be adding results in the next few days. Please refer to W1.1 for further updates
>
> > **Q5**: Can authors compare the proposed method with the method mentioned in reference [1]?
>
> Thank you for the suggestion. We have listed below the performance of [1] when adapted for an unsupervised distillation setting on CIFAR-100. We compute the losses of [1] over the embeddings features. To highlight the differences to our method, [1] applies the normalisation to the features **before** computing inter and intra category similarities. This pre-normalisation has the same effect of L2 normalisation which causes a loss in original manifold information.
>
> | **Method** | **Resnet20/56** | **Resnet8x4/32x4** | **VGG8/13**   | **WRN16/40**  |
> |--------|-------------|----------------|-----------|-----------|
> | DIST[1] | 67.13       |  67.67        | 73.40     | 72.85     |
> | Ours   | **71.11**   | **73.90**      | **74.58** | **74.65**     |
>
>
> > **Q6**: Can the authors provide additional details about ViT training in Tables 1 and 6? Were all ViTs trained from scratch?
>
> Yes, all (student) models (including ViTs) are trained from scratch. Teacher models are obtained off-the-shelf (more details in SEED). For ImageNet, ViT-tiny is the smallest network in the family. We use the patch_size=12, embedding dimension=192, depth=12, num_heads=3. This architecture definition is also supported by the popular library Timm [3]. For CIFAR-100, we scaled down the teacher and student networks to account for the relatively smaller dataset and image size of CIFAR-100. For CIFAR-100, the ViTs have patch size of 8, embedding size=384, num_heads = 12 for teacher and 3 for student and depth=7 for teacher 3 for student. They are trained following the hyper-params as used for other ImageNet and CIFAR training respectively. We have added this information to the appendix as well. Also, we will be sharing the official repository upon acceptance to reproduce **all** of our results including pre-trained models.
>
> References:
>
> [1] Huang, T., You, S., Wang, F., Qian, C., & Xu, C. (2022). Knowledge distillation from a stronger teacher. Advances in Neural Information Processing Systems, 35, 33716-33727.
>
> [2] Liu, H., & Ye, M. (2022, October). Improving Self-supervised Lightweight Model Learning via Hard-Aware Metric Distillation. In European Conference on Computer Vision (pp. 295-311). Cham: Springer Nature Switzerland.
>
> [3]: Ross Whitman (2019). Pytorch Image Models. https://github.com/huggingface/pytorch-image-models

---

> ### Author Response · Authors · 2023-11-19
> **Comparisons with SMD (ImageNet experiment) (3/2)**
>
> In addition to the CIFAR-100 evaluation (provided in the previous response), we share below results for the task of unsupervised distillation from the moco-v2 teacher (from our and other baseline's setting) to a ResNet-18 utilising the official implementation of SMD. We used SMD's official source code with official hyper-parameters for this experiment.
>
> |      |   Top-1   |   KNN-10  |
> |------|:---------:|:---------:|
> | SMD  | **59.56** | 49.69     |
> | Ours | 59.24     | **55.04** |
>
> The results indicate an expected trend as we have seen in the case of other baselines. We are competitive in terms of top-1 with SMD but perform substantially better on the KNN metric.
>
> **Transfer Learning**
>
> Here, we report the transfer learning performance of SMD compared to ours. To recall, we learn the classification hyper-parameters for each model on each dataset independently. We observe that the CoSS student outperforms SMD by a large margin.
>
> |      |  CIFAR-10 | CIFAR-100 | STL-10    | Caltech-101 | Pets      | Flowers   | DTD       |
> |------|:---------:|:---------:|-----------|-------------|-----------|-----------|-----------|
> | SMD  | 86.47     | 64.42     | 94.24     | 80.59       | 74.59     | 78.97     | 69.31     |
> | Ours | **89.84** | **70.03** | **95.31** | **87.06**   | **80.31** | **87.04** | **71.54** |
>
> **OOD Robustness**
>
> For brevity, we report consolidated results of the OOD experiments. For a detailed breakdown, we request the reviewer to view the (soon to be uploaded) new version of the paper.
>
> |      | ImageNet-v2 | ImageNet-S | ImageNet-C |
> |------|:-----------:|:----------:|------------|
> | SMD  | 44.38       | 10.27      | 33.89      |
> | Ours | **48.67**   | **12.85**  | **39.81**  |
>
> **Image Segmentation**
>
> |      | CamVid (Acc$_p$) | CamVid (IoU$_m$) | Cityscapes (Acc$_p$) | Cityscapes (IoU$_m$) |
> |------|:----------------:|:----------------:|----------------------|----------------------|
> | SMD  | 75.67            | 0.1620           | 82.38                | 0.2774               |
> | Ours | **88.00**        | **0.2855**       | **84.39**            | **0.3115**           |
>
> For the SMD distillation on the remaining architectures, we have started the trainings but unfortunately they will not be finishing before the rebuttal deadline. We will be adding them to our work upon completion.
>
> **Lastly, we also would like to sincerely thank the reviewer for their suggestions which has allowed us to strengthen our manuscript.**

---

> ### Comment · Reviewer_sCnR · 2023-11-20
> **Official Comment by Reviewer sCnR**
>
> Thanks to the author for their engagement in the ablation study.
> There is still a lack of the ablation study on $L_{ss}$ and $L_{co}$. Can authors conduct the experiment without $L_{co}$?
>
> For the comparison with DIST, Did you use Pearson’s distance for DIST?

---

> > ### Author Response · Authors · 2023-11-20
> > **Author Response**
> >
> > > Can authors conduct the experiment without $L_{co}$?
> >
> > Below we compare results with only individual components of the loss.
> >
> > |              | $L_{co}$ | $L_{ss}$ | $L_{coss}$ |
> > |--------------|:--------:|:--------:|------------|
> > | Resnet8x/32x |   72.05  |   73.53  | **73.90**  |
> > | Resnet-20/56 | 70.58    | 70.42    | **71.11**  |
> >
> > > For the comparison with DIST, Did you use Pearson’s distance for DIST?
> >
> > Yes, we utilised the Pearson's distance for DIST.

---

> > ### Author Response · Authors · 2023-11-23
> > **Thank you!**
> >
> > **As we approach the end of this discussion phase, we express our gratitude to the reviewer for their insightful critique. We kindly invite the reviewer to evaluate the revised manuscript, thoughtfully refined to integrate their recommended modifications.**

---

### Official Review · Reviewer_VMp2 · 2023-10-30

**Soundness:** 2 fair
**Presentation:** 3 good
**Contribution:** 2 fair
**Rating:** 6
**Confidence:** 3

**Summary:**

This work proposes the CoSS for efficient unsupervised knowledge distillation. Previous works rely on a large feature queue to compute the teacher knowledge, which consumes large memory and computation. CoSS can perform unsuperivsed knowledge distillation on a mini-batch. Specifically, they extract the embedding from the penultimate layer of the network to form a embedding matrix. Then, CoSS minimizes the feature similarity and space similarity between teacher and student. Experiments on various downstream tasks and backbones showcases its performance.

**Strengths:**

1. The proposed method uses a smaller embedding queue for unsupervised KD.
2. The designed loss, feature similarity and space similarity, is easy to follow.
3. The authors conduct extensive experiments to validate their method.

**Weaknesses:**

1. The proposed feature similarity and space similarity is neither well-explained nor intuitive. The authors discuss the reason why they apply normalization on the embedding matrix and use the cosine similiaity instead of the L2 distance in Section 4. The author *treat the embedded manifold as a topological manifold*, and then introduce an argument based on Homeomorphism. Such a conclusion **assumes** that the unsupervised learning methods learn a low dimensional manifold and the manifold is locally euclidean. However, there is a lack of references or theoretical analysis to support their point.
2. The designed loss is analogous to contrastive learning, which computes the cosine similarity between two normalized feature. However,
the discussion does not explain why they only consider the positive samples while neglect the negative samples. I think this may be the key difference from other methods.

**Questions:**

1. Although the method is designed for unsupervised KD, it seems that the method can be used for supervised KD based on their argument.
2. It is unclear why the introduced losses discard the negative samples.
3. In the comparison to CRD, I guess the authors implement the negative contrastive learning. What is the performance of CRD when negative samples are removed?
4. Also, for other comparison method, what is the performance if the negative samples are removed?
5. What is the performance of the proposed model if adding negative samples?
6. The paper claims efficiency as their advantage. Is it possible to improve the performance using a larger batch size?
7. Please add more discussion regarding Section 4.

---

> ### Author Response · Authors · 2023-11-16
> **Author Response (1/2)**
>
> **We thank the reviewer for recognising the effort we put into conducting extensive experiments. Their acknowledgment of our work is truly appreciated. We value the opportunity to engage in a meaningful discussion about our submission, and the reviewer’s comments contribute significantly to that dialogue.**
>
> **We will be uploading the new version of our work once we address the concerns and suggestions of all reviewers. We appreciate the reviewer’s patience in this regard.**
>
> > **W1.A**: The proposed feature similarity and space similarity is neither well-explained nor intuitive. The authors discuss the reason why they apply normalisation on the embedding matrix and use the cosine similarity instead of the L2 distance in Section 4
>
> In section 4, we highlight that normalising the features (L2) alone erases the information present in the pre-normalization embedding manifold. A simple argument for this we presented is that all points lying on a ray through the origin will coincide on the hypersphere. With these collapsed representations we cannot model the teacher's original (unnormalized) manifold which was our original goal. For this purpose, we introduce space similarity. For two points now coinciding on the d-hypersphere will have different contributions to the space similarity loss. The student will learn how data is distributed along different dimensions (up to some scalar). We provide a toy example to help visualise this aspect. In the figure 3 of the paper, we showed that the baseline SEED, though it learns well separated embeddings, is not able to capture the teacher’s embedding local shape as effectively as SEED.
>
> Lastly, we hope to clarify that for (unit) normalised features, L2 distance and cosine similarity capture similar information. We do not prefer one over the other in our formulation. This can be demonstrated by the following derivation. The Euclidean distance between two vectors $\mathbf{u}$ and $\mathbf{v}$ is given by:
> $$D_{\text{Euclidean}}(\mathbf{u}, \mathbf{v}) = \|\mathbf{u} - \mathbf{v}\|_2 $$
>
> For normalized vectors, we have:
> $$ \|\mathbf{u} - \mathbf{v}\|_2 = \sqrt{(\mathbf{u} - \mathbf{v})^T (\mathbf{u} - \mathbf{v})} $$
> $$ = \sqrt{\|\mathbf{u}\|_2^2 - 2 \mathbf{u}^T \mathbf{v} + \|\mathbf{v}\|_2^2} $$
> $$= \sqrt{2 - 2 \cos(\theta)} $$
>
> Here, $\theta$ is the angle between the two vectors which is also captured by cosine similarity. Hence, we can either aim to minimise the distance or maximise the cosine similarity.
>
> > **W1.B**: The author treat the embedded manifold as a topological manifold, and then introduce an argument based on Homeomorphism. Such a conclusion assumes that the unsupervised learning methods learn a low dimensional manifold and the manifold is locally euclidean. However, there is a lack of references or theoretical analysis to support their point.
>
> Thank you for highlighting this issue. Many manifold learning techniques compute local neighbourhood distance with the assumption that the space is locally euclidean[1,2,3]. Explicit statements about the assumption of a locally Euclidean manifold in the context of unsupervised learning can be relatively rare, as this assumption is often fundamental to the underlying methods without being explicitly articulated in the papers. For example, many unsupervised learning methods employ manifold learning based data visualisations which implies that the learnt manifold is locally euclidean [4,5]. We have now added this discussion to Section 4 to strengthen our assumption.
>
> > **W2**: The designed loss is analogous to contrastive learning, which computes the cosine similarity between two normalised features. However, the discussion does not explain why they only consider the positive samples while neglecting the negative samples. I think this may be the key difference from other methods.
>
> Thank you for providing valuable insights. As noted by the reviewer, our methodology markedly deviates from baselines employing a contrastive learning framework for distillation. While we appreciate the acknowledgment of this distinction, we respectfully disagree with the assertion that our alignment objectives closely resemble conventional contrastive learning techniques. Rather, our approach shares greater affinity with methods which in the reviewer's categorization, exclusively relies on only positive samples [6,7]. An advantage of focusing on positive samples lies in the observation that comparisons with negative samples (as done by the baselines) lead to increase in training time and GPU compute. In summary, our objective focusing only on positive samples is not only more effective, but also more efficient, particularly in scenarios where resources may be constrained.

---

> ### Author Response · Authors · 2023-11-16
> **Author Response (2/2)**
>
> > **Q1**: Although the method is designed for unsupervised KD, it seems that the method can be used for supervised KD based on their argument.
>
> Yes, our method can be augmented with any existing supervised learning method. We mentioned this possibility in the discussion section.
>
> > **Q2**: It is unclear why the introduced losses discard the negative samples.
>
> The response is provided in **W2** above.
>
> > **Q3**: In the comparison to CRD, I guess the authors implement the negative contrastive learning. What is the performance of CRD when negative samples are removed?
>
> Yes, we utilised the complete objective of CRD which incorporates positives and negatives in the loss computation. Without the contribution of the negatives, CRD loss failed to optimise in our recent experiments achieving classification score of 1\% equivalent to that of random classification.
>
> > **Q4**: Also, for other comparison methods, what is the performance if the negative samples are removed?
>
> Given that the other methods are also contrastive learning based and removing negatives had a drastic impact on the CRD training. We are currently running a training with SEED’s objective with negative comparisons removed and will provide an update shortly.
>
> > **Q5**: What is the performance of the proposed model if adding negative samples?
>
> As alluded to earlier in our responses, our solution follows an alternative route to that of contrastive learning. We cannot add negatives (samples to increase dissimilarity from)  in the space similarity as this would change the entire paradigm under which we operate. Our intention is not to increase dissimilarity between positive-negative dimensions which is the core idea of contrastive learning.
>
> > **Q6**: The paper claims efficiency as their advantage. Is it possible to improve the performance using a larger batch size?
>
> The ablation over the batch size does not indicate that the performance can be increased by increasing the batch size directly. Please note that we did not perform any hyperparameter optimisation for this study.
>
> | **Method** | $b=32$ | $b=64$ | $b=128$   | $b=256$  | $b=512$ | $b=1024$ |
> |--------|-------------|----------------|-----------|-----------|-----------|-----------|
> | Resnet8x4/32x4   |   73.21    |  73.62        |  73.90     | 73.35     | 72.57 |  72.09  |
> | Resnet20/56   |   70.34    |  71.32        |  71.11     | 71.05     | 70.59 | 70.11
>
> > **Q7**: Please add more discussion regarding Section 4.
>
> Thank you for the suggestion. Based on the discussions highlighted in the review, we have added more discussions in section 4 to support our assumptions and intuition. We have also included details about the setup used for the visualisations in Figure 3.
>
> **References**
>
> [1] van der Maaten, L. & Hinton, G. (2008). “Visualizing Data using t-SNE” . Journal of Machine Learning Research, 9, 2579--2605.
>
> [2] Cayton, Lawrence. “Algorithms for manifold learning”. eScholarship, University of California, 2008.
>
> [3] Hinton, Geoffrey E., and Sam Roweis. "Stochastic neighbor embedding." Advances in neural information processing systems 15 (2002).
>
> [4] Oord, A. V. D., Li, Y., & Vinyals, O. (2018). “Representation learning with contrastive predictive coding”. arXiv preprint arXiv:1807.03748.
>
> [5] Zhuang, Weiming, et al. "Collaborative unsupervised visual representation learning from decentralized data." Proceedings of the IEEE/CVF international conference on computer vision. 2021.
>
> [6] Jean-Bastien Grill, Florian Strub, Florent Altché, Corentin Tallec, Pierre H Richemond, et al.. Bootstrap Your Own Latent: A new approach to self-supervised learning. Neural Information Processing Systems, 2020.
>
> [7] Xinlei Chen and Kaiming He. Exploring Simple Siamese Representation Learning. CVPR (2021).

---

> > ### Author Response · Authors · 2023-11-20
> > **Response Q4. Continued.**
> >
> > > Q4: Also, for other comparison methods, what is the performance if the negative samples are removed?
> >
> > SEED, computes similarity over a feature queue(size $k$) which consists of negative samples and the positive sample corresponding to the teacher output. It applies softmax to the computed similarity scores. This step in the absence of negatives simply assigns 1 to remaining positive similarity scores (for both teacher and student). Similar to what we observed for CRD, the training fails due to lack of informative signals.

---

> ### Comment · Reviewer_VMp2 · 2023-11-21
>
> Dear authors,
>
> Thanks four your reply.
>
> - The clarification about positive samples is important to understand the difference from other constrative learning methods. Here is a reference [1] may be helpful to you.
>
> - Regarding Sec. 4, it would be helpful if you add these discussions.
>
> I have a concern that in Sec. 5.7 (page 8), the authors claim that *''The authors
> highlighted that CRD can be used for unsupervised scenario, however, we find its performance to be
> lacking compared to CoSS''*. However, I note that the authors try to prove this argument in a limited scenarior as described in **Methodology**, which is not rigorous.
>
> Regards,
>
>
> [1] Understanding self-supervised learning dynamics without contrastive pairs. Yuandong Tian · Xinlei Chen · Surya Ganguli. ICML 2021

---

> > ### Author Response · Authors · 2023-11-21
> > **Author Response**
> >
> > We thank the reviewer for their feedback and engagement in the discussion process.
> >
> > > **The clarification about positive samples is important to understand the difference from other contrastive learning methods. Here is a reference [1] that may be helpful to you.**
> >
> > We thank the reviewer for sharing the reference. We have taken it into account to enrich our text in section 4 and section 6.
> >
> > > **Regarding Sec. 4, it would be helpful if you add these discussions**
> >
> > Reviewing [1] and reviewer feedback, we have added further discussion in light of SSL training. Similar to [1], we believe that the absence of contrastive objective in our work motivates the learning network to be similar to the target (teacher) network. However, one key difference which we will like to highlight is the difference in problem setting which impedes drawing conclusions about our training from [1]. SSL training and SSL objective for unsupervised distillation are significantly different problems. SSL training (those considered in [1]) update parameters of all networks involved in the training. However, incase of unsupervised distillation the teacher weights are required to be frozen from start till the end of the training. Moreover, what we also observed was that a predictor was not required by a student. Our CIFAR-100 experiments (prev. section 5.7 and now section 5.8) were conducted without a projection-head/predictor. This observation goes against the findings of [1] which emphasised on its importance. Based on this evidence, we believe further study specific to an unsupervised distillation is required to draw new observations/conclusions.
> >
> > The above discussion is added to Section 6 (Discussion) apart from highlighting the positive alignment aspect in section 4.
> >
> > > **I have a concern that in Sec. 5.7 (page 8), the authors claim that ''The authors highlighted that CRD can be used for unsupervised scenario, however, we find its performance to be lacking compared to CoSS''. However, I note that the authors try to prove this argument in a limited scenario as described in Methodology, which is not rigorous.**
> >
> > Thank you for your follow-up query. We followed CRD's sampling strategy without labels, which is to consider samples from random indices $i\neq j$ as negatives for $x_i$. This approach is described in their section 4.4 of the paper. It is important to note the values in Table 6. of CRD utilises the ground-truth Classification objective under two different sampling strategies. Removing the supervision, which is what we implemented for comparison, the performance drops further. We have now removed the specific remark about CRD’s drop in performance.
> >
> > **We will soon upload the revised draft for the reviewer's assessment.**

---

> > > ### Comment · Reviewer_VMp2 · 2023-11-22
> > >
> > > Dear authors,
> > >
> > > Thanks for your response.
> > >
> > > After reading the rebuttal, I think it is safe to improve the rating.
> > >
> > > The authors adress most of my concerns and correspondingly provide a revised manuscript.
> > >
> > > Although Reviewer sCnR thinks the formulation of the proposed loss is similiar with other work, which is also my concern at another point, I think the novelty is that CoSS proposes to preserve the learned manifold of the teacher network.
> > >
> > > Regards,

---

> > > > ### Author Response · Authors · 2023-11-22
> > > > **Thank you!**
> > > >
> > > > Thank you for your thoughtful reconsideration and acknowledgment of our efforts in addressing your concerns. We appreciate your willingness to revise the rating accordingly.

---

### Official Review · Reviewer_ahHP · 2023-10-31

**Soundness:** 3 good
**Presentation:** 2 fair
**Contribution:** 2 fair
**Rating:** 6
**Confidence:** 4

**Summary:**

This paper is about knowledge distillation without ground-truth labels. A method named CoSS is proposed. A loss based on space similarity loss is introduced alongside with normalized cosine similarity. Specially, each dimension of the student feature space is required be similar to the corresponding dimension of the teacher. Experiments are done to compare with other methods on computer vision tasks.

**Strengths:**

1. The CoSS method is simple and does not require importance sampling.
2. The paper is in general well to follow.

**Weaknesses:**

Please check the questions part.
1. Comparison over baseline is lacked.
2. Performance on CNN and ViT needs more reasonable analysis.
3. Sensitivity of hyperparameter.
4. More details on the computational efficiency are required.

**Questions:**

1. The space similarity is like the traditional cosine similarity loss. But the comparison over the cosine similarity is lacked. It is difficult to evaluate the effects of so-called space similarity.
2. CoSS performs worse on CNN->CNN distillation (Table 1). Could the authors provide more analysis on why the method works well on ViT, but not so good on CNNs?
3. The hyperparameter of lambda is somewhat sensitive to different architectures and datasets. How to choose the appropriate lambda needs further discussion. If hyperparameter search is required, additional training cost is required.
4. The authors claims that the CoSS is faster and more efficient. Yet the comparative details on computational efficiency is not provided. The analysis on how the method is of high efficiency is also lacked.

---

> ### Author Response · Authors · 2023-11-16
> **Author Response (1/1)**
>
> **We’d like to thank the reviewer for dedicating their valuable time to review our work. In particular, we are pleased to find that the reviewer appreciated the simplicity and presentation of our work. They have raised interesting points, which we aim to address below.**
>
> **We will be uploading the new version of our paper once we address the concerns and suggestions of all reviewers. We appreciate the reviewer’s patience in this regard.**
>
> > **Q1 & W1**: The space similarity is like the traditional cosine similarity loss. But the comparison over the cosine similarity is lacked. It is difficult to evaluate the effects of so-called space similarity.
>
> Thank you for the suggestion. We would like to take this opportunity to highlight that we provide an ablation study over $\lambda$ in section A2.1. $\lambda=0$ represents the case of only cosine similarity of features. New experiments on ImageNet ResNet-18 & Eff-b0 with ($\lambda=0$) reveal that we gain roughly $0.9\%$ and $0.8\%$ top-1 by employing space similarity over the baseline. We have now added these results to the appendix.
>
> > **Q2 & W2**: CoSS performs worse on CNN->CNN distillation (Table 1). Could the authors provide more analysis on why the method works well on ViT, but not so good on CNNs?
>
> Thanks for raising this interesting perspective. Underperforming due to differences in architecture is a widely observed phenomenon in the context of supervised distillation [1]. A reason for the drop is owed to different inductive biases present in the different teacher-student architectures. Contrative objectives encourage the model to learn discriminative features. W.r.t distillation, contrastive objectives encourage the student to learn discriminative features as learnt by their teachers. Given different network capacities and inductive biases, we suspect that these are stronger conditions than our alignment based objective.
>
> > **Q3 & W3**: The hyperparameter of lambda is somewhat sensitive to different architectures and datasets. How to choose the appropriate lambda needs further discussion. If hyperparameter search is required, additional training cost is required.
>
> Thank you for the insightful suggestion. Indeed, ideal value for $\lambda$ varies across architectures. We suspect that this is due to differing inductive biases of these architectures. In order to choose the best $\lambda$ for each student-teacher pair, unfortunately, one has to evaluate multiple configurations. However, based on our experiments, $\lambda=0.5$ provides strong results which also extends readily to ImageNet like datasets and for different architectures (Resnet, EfficientNets, ViTs). In addition, we hope to highlight that the additional cost of hyperparameter search is pertinent to all methods. But, given the speed advantage that our method possesses, this search will be relatively faster than baselines’.
>
> > **Q4 & W4**: The authors claims that the CoSS is faster and more efficient. Yet the comparative details on computational efficiency is not provided. The analysis on how the method is of high efficiency is also lacked.
>
> Thank you for the feedback. We have now also added the table reporting the absolute values for GPU utilisation and training time per batch to the appendix. They reflect the values presented in Fig 1 of our paper. Moreover, we have also added relative speed and GPU utilisation (compared with CoSS) to Table 1 as well.
>
> **EDIT:** We will also be reporting total training resources, which is (how much GPU memory is consumed by a method) * (how long the method takes to complete the training). Alongside we will also report the trend of memory and time consumption over the course of training in our paper.
>
>  **References**:
>
> [1] Tian, Yonglong, Dilip Krishnan, and Phillip Isola. "Contrastive Representation Distillation." In International Conference on Learning Representations. 2019.

---

### Official Review · Reviewer_jpJi · 2023-11-01

**Soundness:** 2 fair
**Presentation:** 2 fair
**Contribution:** 3 good
**Rating:** 5
**Confidence:** 3

**Summary:**

This paper tackles self-supervised distillation, specifically targeting the challenge of existing methods requiring extensive sample queues. To address this, the authors have innovatively introduced a loss based on dimension-specific spatial similarity. The novel CoSS supervision framework is composed of conventional sample-based similarities in conjunction with space similarities, thereby effectively emulating the semantic and structural attributes of the data manifold as captured by teacher models. Many experimental validations demonstrate that CoSS not only achieves performance on par with existing methods but also with enhanced efficiency.

**Strengths:**

1. The math formulation of the proposed space similarity is elegantly concise, suggesting an inherent capability to capture and learn the manifold's structure effectively.

2. The diversity of experimental settings presented allows readers to gain a thorough understanding of the proposed CoSS loss's capabilities and performance.

3. The organization of this paper benefits from a clear and logical progression that facilitates comprehension of the material presented.

**Weaknesses:**

1. The Methods section could be enhanced by incorporating a simplified example or illustrative figure to show the concept of space similarity.

2. The discussion on topological spaces in Section 4 is commendable. However, It should be noted that L2 normalization in previous approaches is designed to conform the semantic manifold to a hyperspherical space, thereby constraining the metric within the bounds of the cosine similarity. That means the manifold of teacher feature space is already a cosine space and the similarity is determined by the inner product between two hyper-sphere spaces. In light of this, the paper would benefit from a rigorous comparison demonstrating the superiority, if any, of a d-dimensional Euclidean manifold over a d-dimensional hypersphere for the learning tasks at hand.

**Questions:**

1. This paper could be strengthened by investigating the connection between batch normalization and the proposed CoSS. Can the authors provide insights or very simple results about how BN might influence or interact with CoSS?

2. In my view, to some degree, the joint constraint in CoSS appears to share conceptual relations with the optimal transport (Sinkhorn function). Could the authors elaborate on any theoretical underpinnings or empirical evidence that supports this connection?

---

> ### Author Response · Authors · 2023-11-19
> **Author Response**
>
> **We extend our sincere gratitude to the reviewer for investing their valuable time and expertise in reviewing our work. We're thrilled to hear their positive assessment of our work, particularly the elegant formulation of the proposed space similarity and the diverse experimental settings presented. We value the reviewer’s feedback and have carefully considered each suggestion to enhance the quality of our manuscript.**
>
> **We will be uploading the new version of our paper once we address the concerns and suggestions of all reviewers. We appreciate the reviewer’s patience in this regard.**
>
> > **W1:** The Methods section could be enhanced by incorporating a simplified example or illustrative figure to show the concept of space similarity.
>
> Thank you for the suggestion. Currently, to enhance clarity on the concept of feature and space similarity, we've included an illustrative figure in Figure 2. We've now incorporated a reference to this figure in our method section to more effectively convey our ideas.
>
> > **W2:** The discussion on topological spaces in Section 4 is commendable. However, It should be noted that L2 normalization in previous approaches is designed to conform the semantic manifold to a hyperspherical space, thereby constraining the metric within the bounds of the cosine similarity. That means the manifold of the teacher feature space is already a cosine space and the similarity is determined by the inner product between two hyper-sphere spaces. In light of this, the paper would benefit from a rigorous comparison demonstrating the superiority, if any, of a d-dimensional Euclidean manifold over a d-dimensional hypersphere for the learning tasks at hand.
>
> Thank you for pointing this out. As highlighted by the reviewer, similarity matching using only the feature happens in the d-hypersphere with projected embeddings from the student and teachers. We address the issue of information loss from the teacher's unnormalized manifold to a d-hypersphere. This matching on the d-hypersphere does not account for the structure of the original unnormalized manifolds of teacher and student. Through our empirical evaluations presented in the paper, we aimed to highlight the benefit of capturing the original teacher’s manifold information. CoSS is able to yield competitive results, moreover, lagging in top-1 does not hinder its performance on robustness and various transfer learning tasks.
>
> > **Q1:** This paper could be strengthened by investigating the connection between batch normalization and the proposed CoSS. Can the authors provide insights or very simple results about how BN might influence or interact with CoSS?
>
> We thank the reviewer for providing an interesting perspective. Indeed, we can draw a connection between Batch Normalisation (BN) [1] and our final objective.
>
> To recall, during training, BN operates on a batch of input data, $X \in R^{b\times d}$, where batch size is $b$ and $d$ is the feature dimension. It first performs standardisation along each spatial (channel) dimension. $\hat{X_{:,I}} = \frac{X_{i} - \mu_i}{ \sigma_i}$
>
> where, $\mu_i$, $\sigma_i$ are the mean and variances respectively for the $i^{th}$ spatial dimension. The normalized values are then scaled by trainable parameters $\gamma$ and $\beta$ as: $Z_{:,i} = \gamma_i \hat{X}_{:,i} + \beta_i$
>
> Here, $\gamma \in R^d$ and $\beta \in R^d$ operate on each spatial dimension. These can be interpreted as affine transformations working on each dimension. We can potentially utilise it to map the standardised student embeddings to the teacher's un-normalized embedding space. The corresponding loss can be defined as follows: $\mathcal{L}= \frac{1}{b}\sum_{i=0}^{i<b}\mathcal{D}(Z^s_i, X^t_i) $
>
> where, $X^t_i$ is the teacher's embedding for the $i^{th}$ sample and $Z^s_i$ corresponds to the BatchNormalized student's embeddings for the same sample. Below we share empirical findings with the BatchNorm formulation on CIFAR-100.
>
> | **Method** | **VGG8/13** | **Resnet8x/32x** | **WRN16/40**  |
> |--------|-------------|----------------|-----------|
> | BN         |  74.01      | 72.22            |  73.42         |
> | Ours       | 74.58       | 73.90            | 74.65         |
>
> > **Q2:** In my view, to some degree, the joint constraint in CoSS appears to share conceptual relations with the optimal transport (Sinkhorn function). Could the authors elaborate on any theoretical underpinnings or empirical evidence that supports this connection?
>
> We thank the reviewer for the valuable suggestion. Unfortunately, due to limited familiarity with the topic, we are unable to provide any meaningful insights on the question raised by the reviewer. However, we are willing to explore this direction in the future.
>
> **References:**
>
> [1] Ioffe, S., & Szegedy, C. (2015, June). Batch normalization: Accelerating deep network training by reducing internal covariate shift. In International conference on machine learning (pp. 448-456).

---

> ### Author Response · Authors · 2023-11-22
>
> Thank you for dedicating your time to reviewing our paper and offering valuable insights and suggestions. As today marks the final day of discussion, we're curious if you've had an opportunity to review our responses to your queries. We're more than willing to provide further explanations or address any additional questions you may have. Your feedback is greatly appreciated.

---

> > ### Comment · Reviewer_jpJi · 2023-11-22
> >
> > Thank you for your explanations of my concerns. I don't have any further questions at present. I will consider my final review accordingly.

---

### Author Response · Authors · 2023-11-20
**General Response**

**We would like to thank the reviewers for providing us with valuable feedback. We have attempted to respond to all the issues and concerns raised during the discussion. We have now uploaded the updated version of our paper which should reflect majority of the changes we have covered in our individual responses.**

We would also like to notify that we have corrected an error in the computation of GPU memory and runtime of SEED and CoSS. The discrepancy arose due to the use of mixed precision in these two methods. After disabling extra optimisations in all methods, we still perform better in terms of resources required. The manuscript now reflects the correct values for all.

---

### Meta-Review · Area_Chair_CGkc · 2023-12-08

**Metareview:**

This paper proposes a method for Knowledge Distillation (KD) from a teacher to a student model in an unsupervised setting, eliminating the need of having class labels during distillation. The authors show that applying a penalty term on the similarity between latent dimensions of teacher and student models can be beneficial.

**Strengths**
- The problem tackled is significant due to the limited research in unsupervised KD.
- The authors have diligently tested their algorithm across a range of models and datasets, providing new evidence during the author-reviewer discussion phase.

**Weaknesses**
- The proposed method seems unable to capture the similarity of teacher-student latent spaces when, for example, one is a rotated version of the other.
- Related to the above, the method is not designed to work with spaces of different dimensionalities. In such cases, the authors employ a projection for distillation, but this approach lacks sufficient insights and justifications.
- There are concerns regarding the baseline results, suggesting a potential lack of rigorous validation against existing methods.
- The effectiveness of the proposed method in semi-supervised or supervised settings remains unclear in the current submission.

While the paper has its own merits, it appears that in its present form, it fails to adequately justify its claims, leading to the recommendation against acceptance.

**Justification For Why Not Higher Score:**

The primary contribution of this work is the observation that the similarity between dimensions of a student and teacher latent space, without normalizing against negative pairs, is beneficial in unsupervised knowledge distillation. However, this claim relies predominantly on experimental validation, lacking any theoretical justification or strong insights. During the review process, none of the reviewers champion the paper, with ratings around the borderline reject level, bar one reviewer (score 6, marginally above the acceptance threshold). The paper was discussed thoroughly during the author-reviewer discussion phase and also afterward between the reviewers and myself. We identified a few shortcomings, such as:

1. The algorithm fails to distill effectively when the student's latent space is a permuted version of the teacher's, which would be ideal.
2. The algorithm's inability to distill knowledge across models of differing dimensionality (authors used an extra projection here, but this needs further justification).
3. there are even concerns regarding the results. For example, reviewer u1Un mentioned that the baseline results do not look convincing enough.

Given these concerns, I recommend rejecting this submission.

**Justification For Why Not Lower Score:**

NA

---

### Decision · Program_Chairs · 2024-01-16

Reject